# Targeting the 16S rRNA Gene by Reverse Complement PCR Next-Generation Sequencing: Specific and Sensitive Detection and Identification of Microbes Directly in Clinical Samples

Simone J. C. F. M. Moorlag,[a] Jordy P. M. Coolen,[a] Bart van den Bosch,[a] Elisabeth Hui-Mei Jin,[a] Jochem B. Buil,[a] Heiman F. L. Wertheim,[a] Willem J. G. Melchers[a]

aDepartment of Medical Microbiology, Radboudumc Center for Infectious Diseases, Radboudumc, Nijmegen, The Netherlands

Simone J. C. F. M. Moorlag and Jordy P. M. Coolen contributed equally to the manuscript. As Simone J. C. F. M. Moorlag wrote the first draft of the paper, she is listed first.

**ABSTRACT** The detection and accurate identification of bacterial species in clinical samples are crucial for diagnosis and appropriate antibiotic treatment. To date, sequencing of the 16S rRNA gene has been widely used as a complementary molecular approach when identification by culture fails. The accuracy and sensitivity of this method are highly affected by the selection of the 16S rRNA gene region targeted. In this study, we assessed the clinical utility of 16S rRNA reverse complement PCR (16S RC-PCR), a novel method based on next-generation sequencing (NGS), for the identification of bacterial species. We investigated the performance of 16S RC-PCR on 11 bacterial isolates, 2 polymicrobial community samples, and 59 clinical samples from patients suspected of having a bacterial infection. The results were compared to culture results, if available, and to the results of Sanger sequencing of the 16S rRNA gene (16S Sanger sequencing). By 16S RC-PCR, all bacterial isolates were accurately identified to the species level. Furthermore, in culture-negative clinical samples, the rate of identification increased from 17.1% (7/41) to 46.3% (19/41) when comparing 16S Sanger sequencing to 16S RC-PCR. We conclude that the use of 16S RC-PCR in the clinical setting leads to an increased sensitivity of detection of bacterial pathogens, resulting in a higher number of diagnosed bacterial infections, and thereby can improve patient care.

**IMPORTANCE** The identification of the causative infectious pathogen in patients suspected of having a bacterial infection is essential for diagnosis and the start of appropriate treatment. Over the past 2 decades, molecular diagnostics have improved the ability to detect and identify bacteria. However, novel techniques that can accurately detect and identify bacteria in clinical samples and that can be implemented in clinical diagnostics are needed. Here, we demonstrate the clinical utility of bacterial identification in clinical samples by a novel method called 16S RC-PCR. Using 16S RC-PCR, we reveal a significant increase in the number of clinical samples in which a potentially clinically relevant pathogen is identified compared to the commonly used 16S Sanger method. Moreover, RC-PCR allows automation and is well suited for implementation in a diagnostic laboratory. In conclusion, the implementation of this method as a diagnostic tool is expected to result in an increased number of diagnosed bacterial infections, and in combination with adequate treatment, this could improve clinical outcomes for patients.

**KEYWORDS** 16S rRNA, reverse complement PCR, RC-PCR, bacterial identification, NGS, molecular diagnostics, clinical diagnostics, infectious disease

Address correspondence to Jordy P. M. Coolen, jordy.coolen@radboudumc.nl.

The authors declare no conflict of interest.

Bacterial culture is the gold standard for microbiological diagnosis in clinical samples from patients suspected of having a bacterial infection. However, false-negative cultures may arise in cases of fastidious or uncultivated bacteria or because of the

prior use of antimicrobial therapies. Over the past 2 decades, molecular diagnostics have significantly improved the ability to detect and identify bacteria in clinical samples. Sequencing of the 16S rRNA gene has become the most widely used tool in the routine clinical microbiological laboratory when a bacterial infection is suspected but cultures remain negative or in cases where the species identification of cultured isolates is required and routine tests fail (1).

The 16S rRNA gene, present in all bacteria, is ~1,500 bp long and consists of nine hypervariable regions (V1 to V9) flanked by highly conserved nucleotide sequences (2). Whereas the variable regions are genus or species specific and can therefore be used for bacterial identification (3), the conserved sequences allow PCR amplification using universal PCR primers (4). Substantial variation exists among the different subregions in their abilities to discriminate among species, and the performances of these subregions also vary among different bacterial taxa (5, 6). As a result, various research studies have assessed different primer pairs, spanning different hypervariable regions, for their ability to detect and accurately identify bacteria (4, 6–8). Amplification of the (nearly) full-length 16S rRNA gene enhances species discrimination (5), but the amplification of large fragments (>1,000 bp) might fail in clinical samples containing low loads of bacteria, and sequencing of larger fragments using short-read sequencing technologies is not feasible. Analysis and sequencing of short reads, such as the V4 subregion (~250 bp), are more sensitive but usually do not capture sufficient sequence variations to discriminate accurately between closely related species (9). For several decades, routine clinical microbiology laboratories have performed 16S rRNA gene sequence analysis by targeting one or more hypervariable regions within the 16S rRNA gene, for example, the V1-V2 or V3-V4 hypervariable regions. By sequencing only part of the gene, the discriminatory power for certain genera in clinical samples may be limited, and resolution to the species level is often unfeasible (10, 11). Furthermore, a well-known limitation of Sanger sequencing, the most widely used method for 16S rRNA gene sequencing in clinical laboratories, is the failure of detection in polymicrobial samples (12).

The accurate identification of the causative infectious pathogen in patients is essential for diagnosis, appropriate antimicrobial treatment, and patient management. Therefore, new methods that are both cost-effective and easy to implement into routine clinical diagnostics are needed to improve the detection and accurate identification of bacteria in clinical samples.

Here, we assessed the resolution of bacterial species identification in clinical samples by a novel method based on targeted next-generation sequencing (NGS) using reverse complement PCR (RC-PCR) amplicons targeting the V1-V6 and V9 subregions of the 16S rRNA gene (Fig. 1). RC-PCR integrates the multiplex target enrichment of short amplicons and indexing in a closed-tube system available in a 96-well plate format and has been shown to be an effective and highly sensitive method for DNA profiling in forensic samples (13) and the detection of mutations and variants of severe acute respiratory syndrome coronavirus 2 (SARS-CoV-2) (14). With this approach, the number of handling steps is significantly reduced, resulting in a reduced risk of contamination and less hands-on time compared to the current library preparation protocols required for sequencing. In this study, the identification potential of this method was investigated in both bacterial isolates from cultures and polymicrobial samples. Moreover, we assessed the clinical utility of this method across clinical samples from patients suspected of having a bacterial infection and compared the results to those of our current diagnostic method using Sanger sequencing of the 16S rRNA gene (16S Sanger sequencing).

## RESULTS

**Identification of bacterial isolates by 16S RC-PCR.** To assess the bacterial identification potential of the 16S rRNA gene RC-PCR (16S RC-PCR) method, 11 bacterial isolates and negative-control samples were subjected to 16S RC-PCR in triplicate. As expected, negative-control samples yielded various background species, including *Alteribacillus* sp.,

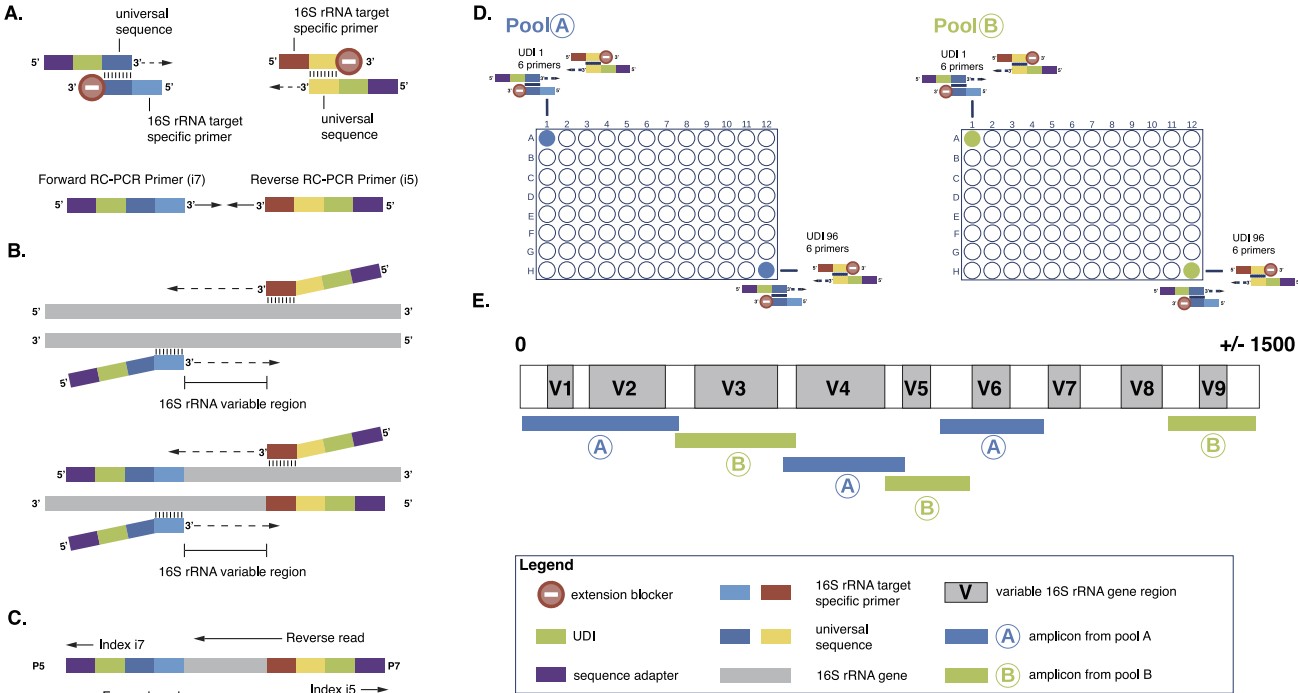

**FIG 1** 16S rRNA gene RC-PCR technology. The schematic is adapted from data reported previously by Kieser et al. (13) and Coolen et al. (14). (A) In the master mix, two types of oligonucleotides are present, one of which contains a unique dual index (UDI), a sequence adapter, and a universal sequence. The second one is the RC primer, which contains an extension blocker, a universal sequence, and a reverse complement of the 16S rRNA gene target. During PCR, after the annealing of the universal sequences, a 16S rRNA gene-specific PCR primer is formed. (B) Regular PCR will be performed. (C) Amplicons will be formed, which are compatible with NGS using the Illumina platform. (D) The RC-PCR is performed on two separate plates, plates A and B. This is to increase sensitivity and minimize chimera formation during PCR. (E) The 16S rRNA RC-PCR design consists of 6 primer pairs covering the V1-V6 and V9 regions of the 16S rRNA gene, covering ±84% of the 16S rRNA gene. See also the supplemental methods and Table S4 in the supplemental material for more details about the design.

*Lepisosteus oculatus*, *Cutibacterium* sp., *Prauserella isguenensis*, *Rubrobacter* sp., and *Staphylococcus capitis* (see Fig. S1 in the supplemental material). As shown in Table S1, all bacterial isolates were accurately identified to the species level for all triplicates. Since various clinical samples referred for 16S rRNA gene sequencing may be polymicrobial, we next evaluated the ability of the 16S RC-PCR method to detect and identify bacteria in polymicrobial samples. One laboratory-derived microbial community made using clinical isolates and a commercial microbial community standard were subjected to 16S RC-PCR in triplicate. All 3 microorganisms in the laboratory-derived microbial community sample in all replicates were accurately identified to the species level (Fig. 2A). In addition, 7 of the 8 bacterial species in the commercial microbial community standard were correctly identified to the species level, and 1 bacterial species was identified correctly to the genus level; it had the highest sequence homology with the *Pseudomonas* metagenome instead of *Pseudomonas aeruginosa* (Fig. 2B). In contrast, Sanger sequencing of the 16S rRNA gene with PCR (16S Sanger) was able to identify only one bacterial species to the genus level in the laboratory-derived microbial community and failed to identify any of the bacterial species in the commercial microbial community standard.

To measure the limit of detection (LOD), both DNA and cell microbial log-distributed standards were subjected to 16S RC-PCR in triplicate (see the supplemental material). The results show that 16S RC-PCR is able to detect *Escherichia coli* with a 16S rRNA abundance of only 0.069% in the log-distributed standard accurately. Based on the dilution series, the LOD is between 47.2 and 4.6 cells for *E. coli* with an abundance of 0.069% (Fig. S2 and Table S2). Furthermore, the 16S RC-PCR method is able to detect the abundance of species comparable to the theoretical 16S rRNA abundance given by ZymoBIOMICS. For the cell standard, we observed an efficiency difference between Gram-positive and -negative species compared to the theoretical 16S rRNA abundance.

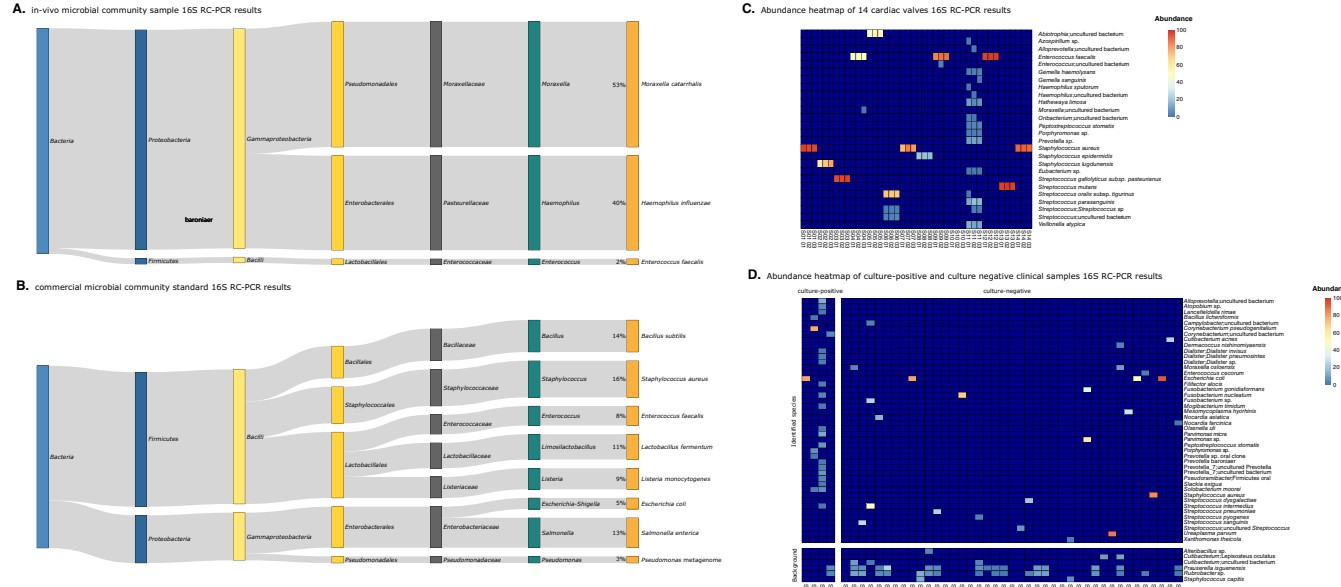

**FIG 2** Representation of RC-PCR results for the mock community and clinical samples. (A and B) Representation of the results for laboratory-derived microbial community samples (A) and the results for the commercial microbial community standard (B). The sizes of the bars are based on the calculated abundances. (C) Abundance heatmap of species detected in 14 heart valves with a positive culture result. (D) Abundance heatmap of culture-positive and culture-negative clinical samples (*n* = 45). Abundance is calculated as the relative percentage of fragments of a given hit compared to all quality-filtered fragments. The added value of the automated taxonomic information retrieved from the RC-PCR Classifier is that it uses the curated SILVA database and taxonomy, enabling straightforward interpretation compared to NCBI BLAST performed using 16S Sanger sequencing.

This did not have an impact on the LOD. It is worth noting that 16S RC-PCR was able to detect *Enterococcus faecalis* 16S rRNA amplicons with an abundance of as low as 0.00067% and an LOD of 0.8 to 0.08 cells; however, these results did not produce enough amplicons to pass quality control (QC) (Fig. S2).

**Comparison of identification potentials between 16S RC-PCR and 16S Sanger sequencing in clinical samples.** To evaluate the ability of the 16S RC-PCR approach to detect species in clinical samples, we performed this method on 14 heart valves from patients suspected of having endocarditis that had been sent to the clinical microbiology laboratory for routine diagnostics and for which a positive culture result was obtained. In 13 of the 14 samples, species identification was obtained and was concordant with the culture results. In 9 of the 14 samples, 16S RC-PCR identified a bacterial organism identical to the results obtained by culture (Table 1 and Fig. 2C). Identification to a more specific taxonomic level was obtained by 16S RC-PCR in 2 additional samples (S03 and S13), whereas the opposite was observed for 1 sample (S05). In one sample, no pathogenic species was detected by 16S RC-PCR (sample S10), whereas culture was positive for *Staphylococcus lugdunensis*. Using the 16S Sanger method, identification failed in 10 of the 14 samples. In 3 out of 14 samples, S01, S03, and S14, the identification of the microorganisms by 16S Sanger sequencing was restricted to the genus level. For one sample (sample S06), identification was improved compared to culture and was identical to the result obtained by 16S RC-PCR.

To further explore the resolution of bacterial species identification by the 16S RC-PCR method, we assessed the performance of this approach across a variety of routinely collected clinical sample types, including culture-negative samples, and compared the results with those of the 16S Sanger method. To this end, we performed 16S RC-PCR on 45 clinical samples that had previously been subjected to 16S Sanger sequencing as part of routine diagnostics. All outcomes were evaluated by a molecular microbiologist and a medical microbiologist, and the results were compared between the two approaches. Species identified in the negative controls that were also present in the clinical samples were excluded from the analysis. Out of the 45 samples, only 4 samples (8.9%) were culture positive (Table 2). By 16S RC-PCR, microorganisms were identified in all 4 culture-positive samples

**TABLE 1** Identification results for 14 heart valves based on culture, 16S Sanger sequencing, and 16S RC-PCR

| Sample | Sample type | Gram stain/culture result(s)[a] | 16S Sanger sequencing result | 16S RC-PCR sequencing result(s) (no. positive/no. of replicates) |
|---|---|---|---|---|
| S01 | Heart valve | NBS, *Staphylococcus aureus* | *Staphylococcus* sp. | *Staphylococcus aureus* (3/3) |
| S02 | Heart valve | GPC, *Staphylococcus lugdunensis* | No identification | *Staphylococcus lugdunensis* (3/3) |
| S03 | Heart valve | GPC, *Streptococcus bovis* group | *Streptococcus* sp. | *Streptococcus gallolyticus* subsp. *pasteurianus* (3/3) |
| S04 | Heart valve | NBS, *Enterococcus faecalis* | No identification | *Enterococcus faecalis* (3/3) |
| S05 | Heart valve | GPC, *Abiotrophia defectiva* | No identification | *Abiotrophia*; uncultured bacterium (3/3) |
| S06 | Heart valve | GPC, *Streptococcus mitis* group | *Streptococcus oralis* subsp. *tigurinus* | *Streptococcus oralis* subsp. *tigurinus* (3/3) |
| S07 | Heart valve | GPC, *Staphylococcus aureus* | No identification | *Staphylococcus aureus* (3/3) |
| S08 | Heart valve | NBS, *Staphylococcus epidermidis* | No identification | *Staphylococcus epidermidis* (3/3) |
| S09 | Heart valve | GPC, *Enterococcus faecalis* | No identification | *Enterococcus faecalis* (3/3) |
| S10 | Heart valve | NBS, *Staphylococcus lugdunensis* | No identification | No identification |
| S11 | Heart valve | NBS, *Streptococcus salivarius* group | No identification | *Streptococcus parasanguinis* (3/3), *Veillonella atypica* (3/3), *Prevotella* sp. (3/3), *Hathewaya limosa* (3/3), *Peptostreptococcus stomatis* (3/3), *Porphyromonas* sp. (3/3), *Gemella haemolysans* (3/3), and other flora of the gut |
| S12 | Heart valve | GPC, *Enterococcus faecalis* | No identification | *Enterococcus faecalis* (3/3) |
| S13 | Heart valve | GPC, *Streptococcus mutans* group | No identification | *Streptococcus mutans* (3/3) |
| S14 | Heart valve | GPC, *Staphylococcus aureus* | *Staphylococcus* sp. | *Staphylococcus aureus* (3/3) |

[a]NBS, no bacteria seen; GPC, Gram-positive cocci.

(100%) (Table 2 and Fig. 2D). In two samples (S16 and S17), 16S RC-PCR identified an increased number of potentially clinically relevant pathogens compared to conventional culture. Using 16S Sanger sequencing, an organism was identified in 1 of the 4 (25%) culture-positive samples (S16), and the identification was concordant with the culture result.

Of the 41 culture-negative clinical samples, bacterial species were detected in 19 samples (46.3%) by 16S RC-PCR, while 16S Sanger sequencing detected bacterial species in 7 samples (17.1%) (Table 3). Among the 7 samples for which 16S Sanger sequencing provided an identification, 16S RC-PCR produced identical results for 2 samples (S33 and S53). Identification to the species level was improved by 16S RC-PCR for 3 samples (S51, S56, and S57), and for 1 sample (S48), an additional pathogen was identified by 16S RC-PCR compared to 16S Sanger sequencing. In sample S36, 16S Sanger sequencing detected the presence of *Sneathia* sp., while this microorganism was not found by 16S RC-PCR, and in sample S48, the microorganism was identified as *Parvimonas micra* by 16S Sanger sequencing and as *Parvimonas* sp. strain KA00067 by 16S RC-PCR.

In all 13 culture-negative samples with a negative 16S Sanger sequencing result, the 16S RC-PCR results were considered potentially clinically relevant as evaluated in a multidisciplinary consultation comprised of a medical microbiologist, a molecular expert, and a bioinformatician. Of these, the identification for 7 samples (S21, S22, S23, S27, S30, S35, and S59) was supported by additional microbiological diagnostic tests that had been performed as part of routine clinical care.

To summarize, for culture-positive (Table 2) and culture-negative (Table 3) samples combined (*n* = 45), bacterial species were identified in 23 samples (51.1%) by 16S RC-PCR and in only 8 samples (17.8%) by 16S Sanger sequencing (Fig. 3A). Identification to the species level was successful for 21 samples (46.7%) using 16S RC-PCR, compared to 3 samples (6.7%) when 16S Sanger sequencing was used. Furthermore, in 4 samples, 16S RC-PCR revealed more than one potential clinically significant species. The number of bacterial species detected by 16S Sanger sequencing was limited to a maximum of 1 species per sample in all samples assessed. In contrast to 16S Sanger sequencing, bacterial species were detected by 16S RC-PCR across all sample types assessed except for bone (Fig. 3B). Remarkably, whereas no organisms were detected in cerebrospinal fluid (CSF) samples by 16S Sanger sequencing, a rate of detection by 16S RC-PCR of 100% (8/8) was found for CSF samples.

## DISCUSSION

Sequencing of the 16S rRNA gene has become a widely used tool in the routine clinical laboratory for bacterial identification in culture-negative clinical samples. In this

**TABLE 2** Comparison of identification results for culture-positive samples using 16S Sanger sequencing and 16S RC-PCR[a]

| Sample | Sample type | Gram stain/culture result(s) | 16S Sanger sequencing result | 16S RC-PCR sequencing result(s) | Working diagnosis | Microbiological result for other samples from the patient |
|---|---|---|---|---|---|---|
| S15 | CSF | GNR, *Escherichia coli* | No identification | *Escherichia coli* | Sepsis | Blood culture positive for *Escherichia coli* |
| S16 | Pus | GNR, *Corynebacterium tuberculostearicum* | *Corynebacterium tuberculostearicum* | *Corynebacterium pseudogenitalium* ATCC 33035, *Porphyromonas sp.*, *Solobacterium moorei; Prevotella sp.* oral clone DA058, *Bacillus licheniformis* | Neck abscess | Negative |
| S17 | Tissue | GNR, GPC, GPR, *Streptococcus anginosus*, *Eikenella corrodens, Prevotella* species, *Parvimonas micra* | No identification | *Alloprevotella*, uncultured bacterium, *Prevotella sp.*, *Parvimonas micra*, *Peptostreptococcus stomatis*, *Solobacterium moorei*, *Atopobium sp.*, *Firmicutes* oral clone, *Fusobacterium nucleatum*, other gut flora | Aortoduodenal fistula | Negative |
| S18 | Tissue | NBS, *Corynebacterium tuberculostearicum*, *Staphylococcus epidermidis* | No identification | *Corynebacterium sp.* | Spleen cyst | Negative |

[a]GPC, Gram-positive cocci; GNR, Gram-negative rods; GPR, Gram-positive rods; CSF, cerebrospinal fluid; NBS, no bacteria seen.

**TABLE 3** Comparison of identification results for culture-negative samples using 16S Sanger sequencing and 16S RC-PCR[a]

| Sample | Sample type | Gram stain/ culture result (s) | 16S Sanger sequencing result | 16S RC-PCR sequencing result(s) | Working diagnosis | Microbiological result(s) for other samples from the patient |
|---|---|---|---|---|---|---|
| S19 | Tissue | NBS, no growth | No identification | No identification | Aortic prosthetic graft infection | Negative |
| S20 | Tissue | NBS, no growth | No identification | *Cutibacterium* sp.[b] | Aortic prosthetic graft infection | Negative |
| S21 | Heart valve | GPC, no growth | No identification | *Streptococcus sanguinis* | Endocarditis | Previous blood cultures positive for *Streptococcus sanguinis* |
| S22 | CSF | NBS, no growth | No identification | *Streptococcus intermedius*, *Fusobacterium* sp., *Cutibacterium* sp.,[b] *Campylobacter* sp. | Multiple brain abscesses | Follow-up CSF sample culture positive for *Streptococcus intermedius*, blood culture positive for *Streptococcus intermedius* |
| S23 | CSF | NBS, no growth | No identification | *Nocardia asiatica* | Meningitis | Previous CSF sample culture positive for *Nocardia* sp. |
| S24 | Heart valve | GPC, no growth | No identification | No identification | Endocarditis | Previous blood cultures positive for *Staphylococcus aureus* |
| S25 | Tissue | NBS, no growth | No identification | No identification | Aortic prosthetic graft infection | Negative |
| S26 | Tissue | NBS, *Candida albicans* | No identification | No identification | Persistent empyema after pneumonectomy for tuberculosis | Negative |
| S27 | CSF | GNR, no growth | No identification | *Escherichia coli* | Meningitis | Previous CSF sample culture positive for *Escherichia coli* |
| S28 | Tissue | NBS, no growth | No identification | No identification, *Staphylococcus capitis*[b] | Aortic prosthetic graft infection | Negative |
| S29 | Joint fluid | NBS, no growth | No identification | No identification | Surgical site infection | Previous blood cultures positive for *Citrobacter koseri* |
| S30 | Heart valve | GPC | No identification | *Streptococcus pneumoniae* | Disseminated pneumococcal sepsis | Previous blood cultures positive for *Streptococcus pneumoniae* |
| S31 | Heart valve | NBS, no growth | No identification | No identification | Endocarditis | Negative |
| S32 | Heart valve | NBS, no growth | No identification | No identification | Endocarditis | Negative |
| S33 | Pus | NBS, no growth | *Fusobacterium nucleatum* subsp. *vincentii* | *Fusobacterium nucleatum* subsp. *vincentii* | Liver abscesses | Negative |
| S34 | Heart valve | NBS, no growth | No identification | No identification | Endocarditis | Negative |
| S35 | CSF | NBS, no growth | No identification | *Streptococcus pyogenes* | Meningitis | Serology positive for recent infection with hemolytic streptococci (ASO + anti-DNase B Ab) |
| S36 | Tissue | GPC, no growth | *Sneathia* sp. | No identification | Aortic prosthetic graft infection | Abscess near vertebra 1 mo later positive for *Enterococcus faecium* |
| S37 | Bone | GPC, no growth | No identification | No identification | Bone scan abnormalities | Negative |
| S38 | Bone | NBS, no growth | No identification | No identification | Spondylodiscitis | Negative |
| S39 | Tissue | NA | No identification | No identification | Sudden infant death syndrome | Negative |
| S40 | CSF | NBS, no growth | No identification | *Streptococcus* sp. | Meningitis | Negative |
| S41 | Tissue | GNR, no growth | No identification | *Streptococcus dysgalactiae* subsp. *equisimilis* | Cellulitis | Negative |
| S42 | Bone | GPR, no growth | No identification | No identification | Osteomyelitis | Not performed |
| S43 | Bone | GPR | No identification | No identification | Discitis | Not performed |
| S44 | Tissue | NA | No identification | No identification | Sudden infant death syndrome | Negative |

**TABLE 3** (Continued)

| Sample | Sample type | Gram stain/ culture result (s) | 16S Sanger sequencing result | 16S RC-PCR sequencing result(s) | Working diagnosis | Microbiological result(s) for other samples from the patient |
|---|---|---|---|---|---|---|
| S45 | Bone | NBS, no growth | No identification | No identification | Spondylodiscitis | Negative |
| S46 | Pus | NBS, no growth | No identification | *Staphylococcus capitis*[b] | Spondylodiscitis with abscesses | Negative |
| S47 | Bone | NBS, no growth | No identification | No identification | Osteomyelitis | Negative |
| S48 | Pus | GPC, no growth | *Parvimonas micra* | *Parvimonas* sp. KA00067, *Fusobacterium gonidiaformans* | Brain abscesses | Follow-up sample culture positive for *Parvimonas* sp. |
| S49 | Joint fluid | NBS, no growth | No identification | No identification | Infection of the hip | Negative |
| S50 | Bone | NBS, no growth | No identification | No identification | Mediastinitis | Negative |
| S51 | Pus | NBS, no growth | *Ureaplasma* sp. | *Ureaplasma parvum* serovar 6 | Spondylodiscitis with abscesses | Negative |
| S52 | CSF | NBS, no growth | No identification | *Dermacoccus nishinomiyaensis* | NA | NA |
| S53 | Tissue | NBS, no growth | *Mycoplasma hyorhinis* | *Mesomycoplasma hyorhinis*[c] | Mycotic aneurysm | Negative |
| S54 | Tissue | NBS, no growth | No identification | *Escherichia coli* | Retroperitoneal mass | Negative |
| S55 | Tissue | NBS, no growth | No identification | *Enterococcus cecorum* | Mycotic aneurysm | Negative |
| S56 | Pus | GPC, no growth | *Staphylococcus* sp. | *Staphylococcus aureus* | Subdural empyema | Negative |
| S57 | Pus | NBS, no growth | *Escherichia* sp. | *Escherichia coli* | Sepsis, focus unknown | Blood culture positive for *Escherichia coli* |
| S58 | Joint fluid | NBS, no growth | No identification | *Cutibacterium acnes* | Prosthetic joint infection | Negative |
| S59 | CSF | NBS, no growth | No identification | *Nocardia farcinica* | Lung infiltrate | Previous CSF sample positive for *Nocardia farcinica* |

[a]GPC, Gram-positive cocci; GNR, Gram-negative rods; GPR, Gram-positive rods; CSF, cerebrospinal fluid; NBS, no bacteria seen; Ab, antibody; ASO, Antistreptolysin O; NA, not applicable.
[b]Associated with human infection but also present in the negative control.
[c]Newer taxonomic name.

study, we investigated a novel method, 16S RC-PCR, for the detection of the bacterial 16S rRNA gene, which is based on amplification by RC-PCR and NGS of the V1-V6 and V9 hypervariable regions. Our results show that 16S RC-PCR accurately identified bacterial species that were found to be the cause of infection based on culture. Using 16S RC-PCR, bacterial species were identified in the majority (92.9%) of culture-positive endocarditis cases, whereas by 16S Sanger sequencing, an identification result was obtained for only 28.6% of the samples. Furthermore, the added value for culture-negative clinical samples is impressive; 16S RC-PCR provided species-level identifications for 43.9% (18/41) of the samples, compared to only 7.3% (3/41) by 16S Sanger sequencing. The failure to detect the 16S rRNA gene in culture-positive samples by 16S

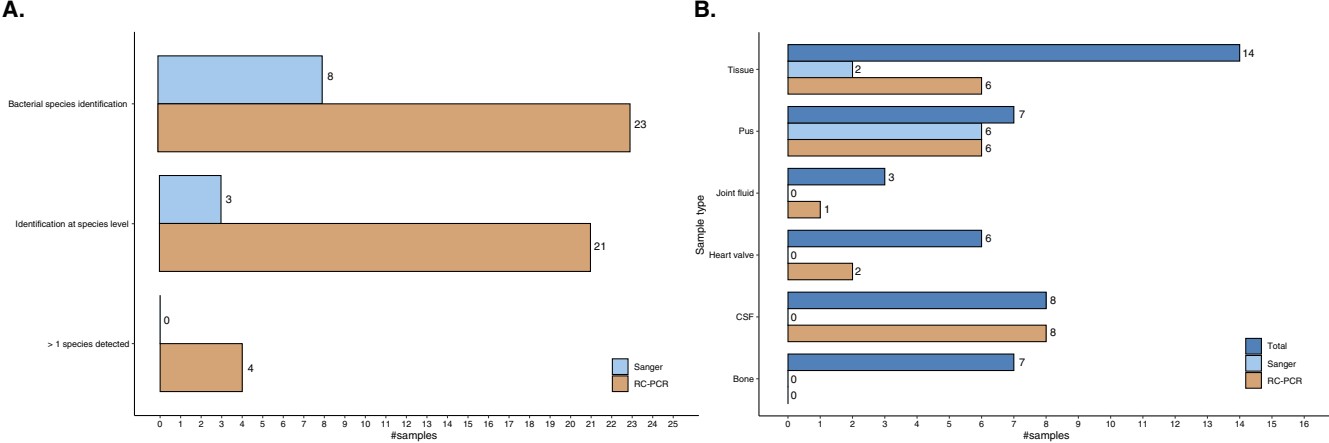

**FIG 3** (A) Comparison of identification results for culture-positive and culture-negative clinical samples using 16S Sanger sequencing and 16S RC-PCR. (B) Comparison of the identification results by 16S Sanger sequencing and 16S RC-PCR for clinical samples (*n* = 45) based on sample type.

Sanger sequencing has been described previously (15, 16) and may be explained by a low number of bacteria present in the sample or sampling error. These data demonstrate the high sensitivity of the 16S RC-PCR method for the identification of bacterial species compared to our currently used 16S Sanger approach.

The detection and accurate identification of bacterial organisms by sequencing of the 16S rRNA are highly affected by the particular region of the 16S rRNA gene targeted (17). The choice of the most optimal primer pair is the subject of ongoing debate and is also dependent on the sequencing platform used (7, 18, 19). Amplicon sequencing of a particular subregion of the 16S rRNA gene (e.g., V1-V2 or V3-V4) is currently the most commonly used strategy in routine clinical diagnostic laboratories. Although the short amplicon sizes ensure sensitivity, this approach is associated with a significant reduction in the precision of species identification. We hypothesized that multiplex target amplification of short amplicons by RC-PCR, covering about 80% of the entire 16S rRNA gene, would improve the identification accuracy and sensitivity in clinical samples that are expected to contain low bacterial loads.

16S Sanger sequencing is known for its inability to identify multiple different bacterial species present in one sample. With the advent of NGS technology, the simultaneous identification of different bacterial species in polymicrobial samples became feasible (20). Here, we show that the NGS-based 16S RC-PCR method could indeed detect and identify the different bacterial species present in our mock communities, whereas 16S Sanger sequencing failed to identify more than one species. Importantly, we also observed the detection of multiple clinically relevant bacterial species by 16S RC-PCR in our clinical samples, and this resulted in the identification of additional species that were not detected by conventional methods. For example, obligate anaerobic bacteria are important pathogens in many types of infections but are difficult to culture and, as a result, are often not identified by conventional culture methods (21). A failure to identify anaerobic bacteria may prevent the start of appropriate antimicrobial therapy and may result in treatment failure. For various clinical samples in our study, strict obligate anaerobes were identified by 16S RC-PCR, whereas culture results remained negative for these species.

Furthermore, 16S RC-PCR improved the identification of bacterial species in culture-negative clinical samples compared to 16S Sanger sequencing. The identified bacterial species were considered potentially clinically significant pathogens after evaluation by a clinical microbiologist. Furthermore, the findings were supported by microbiological test results for other samples from the patient in 53.8% (7/13) of culture-negative samples with a negative 16S Sanger sequencing result. In one of the culture-negative samples (S36), 16S Sanger sequencing identified a bacterial species, whereas no species was detected by 16S RC-PCR. The species detected by 16S Sanger sequencing involved *Sneathia* sp., and the patient was treated accordingly. However, the clinical condition of the patient did not improve, and when a second clinical sample from this patient was subjected to 16S Sanger sequencing, the sample became positive for *Enterococcus* sp. Next, antimicrobial therapy was switched, and the patient recovered. Based on the clinical course of this patient, we conclude that the *Sneathia* sp. isolate might have been a contaminant, even though the negative control in the assay was valid. The occurrence of contaminating bacterial DNA in DNA extraction kits, PCR reagents, and the environment is well known and may bias the interpretation of data obtained using molecular methods (22). The increased sensitivity of the 16S RC-PCR method may be associated with a high rate of detection of contaminants compared to less sensitive methods. This issue is particularly important in the context of analyzing clinical samples derived from normally sterile body sites, such as the CSF, heart valves, and joint fluids, because of their generally low bacterial loads (23). Therefore, negative-control samples should be included in every assay and analyzed to identify the potential introduction of contaminating DNA. In our study, the negative controls contained several contaminants that were in part also detected in our clinical samples, which is consistent with the results of other studies that reported the detection of contaminants in low-biomass samples (23). Most of these species are biologically unexpected and not

clearly associated with human infection and therefore can be regarded as contaminants after evaluation. However, four samples (S20, S22, S28, and S46) contained species that can be clinically significant microorganisms but were also found in at least one negative-control sample. Therefore, their presence in the clinical samples cannot be clearly regarded as contamination by their identification alone. In these cases, a comparison of relative abundances may be helpful to ensure the correct interpretation by using an internal control (24), whereas other reports have suggested the use of spiked samples to determine contamination effects (25). In addition, the *Sneathia* sp. case in our study illustrates the importance of a very careful examination of results within the clinical context in a multidisciplinary consultation, followed by consultation between a medical microbiologist and the clinical team, to avoid misinterpretations and to prevent false-positive results (even when negative controls are included). Furthermore, in two samples, the clinically relevant bacterial species *Staphylococcus aureus* (sample S24) and *Neisseria gonorrhoeae* (S45) were identified with coverages of 29.4% and 26.4%, respectively, which are below our set quality control value of 30%, and for this reason, they were excluded, which underlines the importance of a critical inspection of the obtained results as these species could be of clinical relevance and considered to be communicated to the clinical team.

In addition, accurate identification depends on the quality and completeness of the reference database that is used. For identification, the 16S RC-PCR approach makes use of sequences already available within large public reference databases. In this study, we used SILVA (26); however, this method would also work with the NCBI database (27), for instance. Other proposed methods that aim to improve species discrimination by using alternative marker genes compared to the 16S rRNA gene are considered inferior options (28). However, the use of public 16S rRNA gene databases is known for the bias that they may introduce into the data analysis, and manual evaluation of the identification results is of importance.

In conclusion, we demonstrate that the 16S RC-PCR method accurately identifies bacterial isolates to the species level. Moreover, we show that this method can detect and identify different bacterial species present within polymicrobial samples. Importantly, we reveal that the employment of the 16S RC-PCR approach results in an improved detection of clinically relevant bacteria in clinical samples and increased species discrimination compared to 16S rRNA Sanger sequencing. The simple workflow of the 16S RC-PCR method and the minimal hands-on time are likely to enable implementation in the routine diagnostic clinical laboratory in the short term. A potential next step would be the detection of drug resistance genes by this method in addition to species identification. Another appropriate but more expensive and less sensitive method would be to perform human DNA depletion followed by metagenome sequencing, referred to as clinical metagenomics (29). On the basis of these data, we argue that the use of 16S RC-PCR for routine diagnostics will result in an increased sensitivity of the broad-range molecular detection of bacterial pathogens compared to current molecular methods and thereby has the potential to improve clinical outcomes in cases where bacterial infections are suspected but cultures remain negative.

## MATERIALS AND METHODS

**Ethics.** According to the policy of the Radboud University Medical Center, all patients are informed of the use of residual patient material for anonymous research purposes and can opt out. Only clinical samples from patients who did not opt out were included.

**Samples.** The bacterial isolates used in this study included both laboratory strains and patient isolates (see Table S1 in the supplemental material). A microbial community standard (ZymoBIOMICS; Zymo Research, Irvine, CA, USA) was used to evaluate the capacity of the RC-PCR technology to detect and identify bacteria in a polymicrobial sample (Table S3). Additionally, 14 culture-positive cardiac valves from patients with clinical suspicion of endocarditis were collected (Table 1). All clinical specimens were collected according to standard operating procedures in place at Radboudumc, enhancing sterile sampling as much as possible. The GLIMS (version 9) laboratory information system was used to identify 45 clinical samples that had been subjected to 16S Sanger sequencing between 29 April 2019 and 29 October 2020 as part of routine diagnostics. DNA was extracted from bacterial isolates and clinical samples using MagNA Pure 96 (Roche) according to the manufacturer's instructions. Additional details can be found in the supplemental material.

**16S rRNA Sanger sequencing and 16S rRNA RC-PCR.** 16S rRNA Sanger sequencing was performed on clinical samples in our center according to the protocol listed in the supplemental material.

The 16S rRNA region was amplified by RC-PCR using 6 designed primer pairs divided into two pools to amplify the V1-V2, V4, and V6 (pool A) and V3, V5, and V9 (pool B) subregions of the 16S rRNA gene (Fig. 1 and Table S4). See the supplemental material for further details. The total turnaround time of the RC-PCR method was about 26 h, including 2 h of hands-on time, 6.5 h for RC-PCR, 17 h of sequencing time, and 1 h of analysis. Samples were run in triplicate. Sequencing data were processed using the RC-PCR Classifier. The results were quality assessed, and correctly identified species needed to have a 16S rRNA gene coverage of $\geq$30%, a read count of $\geq$100, an abundance of $\geq$1%, and a $k$-mer alignment (KMA) depth of $\geq$10. Negative-control samples, processed in the same DNA extraction, RC-PCR, and sequence runs, were assessed for background signals. Species present in the negative controls (Fig. S1) and also present in clinical samples were regarded as potential contaminants and were not included in the analysis of the identification results of 16S Sanger sequencing versus RC-PCR. The clinical relevance of the identification results was evaluated by a clinical microbiologist for all samples. The 16S RC-PCR method developed in this study has recently been released as the EasySeq 16S rRNA bacterial identification kit by NimaGen BV, Nijmegen, The Netherlands.

**Data availability.** Sequence data and descriptions are available on Zenodo at https://doi.org/10 .5281/zenodo.7466961.

## SUPPLEMENTAL MATERIAL

Supplemental material is available online only.

**SUPPLEMENTAL FILE 1**, DOCX file, 0.5 MB.

## ACKNOWLEDGMENTS

We thank Walter van der Vliet, Simon van Reijmersdal, and Joop Theelen from NimaGen BV for technical support and material supply. We thank Ellen Koenraad for her help with the experiments.

Methodology, J.P.M.C., S.J.C.F.M.M., E.H.-M.J., and J.B.B. Analysis, J.P.M.C., S.J.C.F.M.M., J.B.B., and W.J.G.M. Writing – Original Draft, S.J.C.F.M.M. Writing – Review & Editing, J.P.M.C., H.F.L.W., and W.J.G.M. Supervision, W.J.G.M.

We declare no conflict of interest. The authors of this paper codeveloped, designed, and optimized the assay together with NimaGen BV prior to the release of the EasySeq product. The authors designed the study and developed the bioinformatics independently from NimaGen BV. All data were analyzed and written down without interference by NimaGen BV.

This research received no specific grant from any funding agency in the public, commercial, or not-for-profit sectors.

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
