## [Reviewer comments · Microbiology Spectrum]

Microbiology Spectrum

Targeting 16S rRNA by reverse complement PCR-next-generation-sequencing: specific and sensitive detection and identification of microbes directly in clinical samples

Simone Moorlag, Jordy Coolen, Bart Bosch, Elisabeth Jin, Jochem Buil, Heiman Wertheim, and Willem Melchers

Corresponding Author(s): Jordy Coolen, Radboudumc

Review Timeline:

Submission Date:	November 3, 2022
Editorial Decision:	January 5, 2023
Revision Received:	April 27, 2023
Editorial Decision:	April 28, 2023
Revision Received:	May 8, 2023
Accepted:	May 9, 2023

Editor: Paul Luethy

Reviewer(s): Disclosure of reviewer identity is with reference to reviewer comments included in decision letter(s). The following individuals involved in review of your submission have agreed to reveal their identity: David C Gaston (Reviewer #1)

Transaction Report:

DOI: <https://doi.org/10.1128/spectrum.04483-22>

January 5, 2023

Dr. Jordy P. M. Coolen
Radboudumc
Medical Microbiology
Geert Grooteplein Zuid 10
Nijmegen 6525 GA
Netherlands

Re: Spectrum04483-22 (Targeting 16S rRNA by reverse complement PCR-next-generation-sequencing: specific and sensitive detection and identification of microbes directly in clinical samples)

Dear Dr. Jordy P. M. Coolen:

Link Not Available

Sincerely,

Paul Luethy

Journals Department
Reviewer comments:

Reviewer #1 (Comments for the Author):

In this well written manuscript the authors present a comparison between two methods of 16S rRNA sequencing for identification of bacterial pathogens from clinical specimens. One method is a traditional Sanger sequencing method, and the comparator is an NGS based system utilizing RC-PCR. The authors retrospectively assessed remnant clinical specimens from multiple tissue sources to demonstrate superior performance of the RC-PCR approach. Notably, in instances where more organisms were identified by the RC-PCR approach, the authors performed thorough review of patient cases in a collaborative manner to determine if the identified organisms could be in fact pathogens. This study is limited by the retrospective nature, single center

design, and relatively small number of specimens evaluated. However, it is overall well done, and provides interesting insight into a new method for direct-from-specimen identification using 16S sequencing. This is an impressive article and is overall well performed.

That said, I have two major comments about this manuscript:

The first relates to the comparison between the two assays. This is not an apples-to-apples comparison. It is more akin to a comparison between an apple and an extravagantly baked apple pie. Meaning, the intention of the two methods is the same (identify potential pathogens using 16S sequencing) but the methods used are drastically different. All methods used in the RC-PCR approach, from specimen extraction to bioinformatics, are markedly more robust than those used for the Sanger approach. The Sanger approach used one forward and two reverse, somewhat antiquated post-amplification clean-up steps (including gel extraction, which is notorious for loss of amplicons), a simple bioinformatic approach (BLAST), and a problematic database (NCBI nucleotide). In comparison, the RC-PCR approach used six primer pairs, efficient magnetic bead post-PCR clean-up, a robust k-mer based bioinformatic approach, and a curated database (SILVA). Accordingly, it is not surprising that the RC-PCR approach was superior. Could the Sanger approach be improved to similar performance as the RC-PCR approach with optimized primers, sample processing steps, bioinformatic analysis, and use of a more robust database?

The authors need to address these differences in technique and the potential implications for their results. Additionally, the authors need to clarify why this specific approach to Sanger sequencing was used as a comparator for the RC-PCR method. If this is the Sanger method used in the institutional microbiology laboratory, that needs to be explicitly stated, and it is an entirely acceptable comparator if so. Otherwise, it seems that a clearly inferior approach was chosen as a comparator, making the study appear somewhat biased from the start.

The second major concern relates to the development and potential marketing of this approach. No conflicts of interest are declared by any authors, yet lines 334-336 state, "The 16S RC-PCR method as developed in this study has recently been released as EasySeq™ 16S rRNA Bacterial ID kit by NimaGen B.V., Nijmegen, The Netherlands". Accordingly, clarity is needed as to how the authors came to study this approach if it was already in development for commercial offering. Do the authors have relationships with NimaGen? Did the authors have access to this product before released as a commercial product and was evaluating performance prior to release? Or, did the authors develop this product that was then purchased by NimaGen for release? These questions need to be clarified to ensure there are indeed no conflicts of interest.

Additional minor comments are as follows:

There is an overall lack of quality control data to demonstrate that the 16S amplicons generated by upstream processes are of similar quality before undergoing Sanger or Illumina based sequencing. Please include information in the methods regarding how the efficiency of gel extraction was assessed in the Sanger method prior to Sanger sequencing. How do you know you had enough amplicon remaining after the extraction steps to even sequence anything?

The sentence spanning lines 83-86 needs to be revised for accuracy, addressing the following points: 1) use of the phrase "often fails" requires a reference to define how often this fails, and 2) "most-commonly used NGS platforms" differs by laboratory, so this should be clarified to describe short-read vs long-read sequencing approaches.

The sentence spanning lines 88-90 needs to be revised for accuracy. A reference is needed to define how "most microbiology laboratories" perform 16S sequencing. Does this statement apply globally, only to European or US labs, or some combination thereof? Additionally, this statements grossly oversimplifies the complex methods that comprise 16S sequencing. It suggests "most microbiology laboratories" perform similar methods, which is incorrect. There is not an accepted standard practice for this identification approach. As such different laboratories use different primers, sequencing methodologies, bioinformatic strategies, and databases.

What is meant by "a single closed tube system" on line 103? This seems to contrast the 96 well plate that is presented in Figure 1.

Line 137: please change "send" to "sent"

Additional information is needed regarding the clinical specimens utilized in this specimen (both the heart valves and the additional specimens). Specifically, where these samples handled with strictly sterile methods prior to evaluation? This is very important to describe given the potential for detection of contaminants that could be misidentified as potential pathogens (separate from those detected by use of appropriate negative controls).

Please elaborate on the professions of those involved in the "multidisciplinary consultation" described on line 173.

Lines 202-203 need to be revised for accuracy for the same reasons mentioned above. Stating the "often used 16S Sanger approach" assumes the same method is used by laboratories performing such testing, but that is not accurate. Laboratories use techniques that differ in all steps of the process and as such cannot be reduced with such simplicity.

Although Panels A and B in Figure 2 are visually pleasing representations of the K-mer analysis (using what seems to be Pavian, or a similar program), more exposition is needed in the body text or the figure legend to explain the benefits of taxonomic identification using this bioinformatic approach. As it stands, it may be difficult for a reader without bioinformatic experience to appreciate the additional information provided by such an approach as compared to simply utilizing Blast and the NCBI nucleotide database (as we performed for the Sanger approach).

Please clarify in the figure legend what is meant by "Abundance" in the heatmaps of Figure 2. Does this represent the relative abundance (percentage) of the given organism in the specimen as compared to all reads, all microbial reads, or something else?

In figure 2 D, do the colored bars next to "Identified species" (orange), "Background" (grey), "Culture positive" (yellow), and "Culture-negative" (green) correspond to data in the figure? If these colored bars do not convey information, then please remove them.

Reviewer #2 (Comments for the Author):

General comments: This manuscript by Moorlag et al. explored the use of 16S rRNA reverse complement polymerase chain reaction (16S RC-PCR) which was a commercially available reagents/method for the identification of bacterial species. The manuscript demonstrated that their approach using EasySeq was more successful in identification of bacteria from culture-negative clinical specimens. The word "reverse" kind of confused this reviewer a bit because this reviewer immediately thought of reverse transcription. There were studies that if extracted nucleic acid went through reverse transcription first before 16S PCR, the sensitivity could be higher since 16S rRNA genes are typically multi copy and living bacterial cells make a lot of rRNA as they grow and divide, and this was what this reviewer thought this study was about. Once this reviewer finally understood the technique, this approach was quite smart because it enabled specificity of amplification to useful variable segments of 16S rRNA gene while including the indexing and adapter attachment at the same time. Although CLIA does not apply to the Netherlands, the authors should clearly demonstrate the performance of the assay, including reproducibility (which this reviewer believed the authors had done some of it), limit of detection (CFU/mL), interfering substances, stability, etc. This reviewer was also curious as to how different types of samples were processed. For example, a heart valve or tissue may be eluted and/or homogenized in sterile saline for culture. Was the exact same processed materials used for sequencing and was it done immediately in parallel (culture, Sanger, 16S RC-PCR)? The authors mentioned that sample was processed according to their SOPs but this reviewer was just curious about the workflow. Other comments are below.

Comments

Line 60-61: It should be clarified that increased number of bacterial infections won't necessarily result in better treatment outcomes. It depends on whether the assay truly identifies whether an infection is the cause of the illness and if so, which organism(s) are the etiologic agent(s).

Lines 84-85: It was unclear to this reviewer why this could not be feasible with most commonly used NGS platforms. Please clarify why and which platforms the authors were talking about.

Line 86: which platform is that?

Line 124: please explain what they authors meant by "in vivo"

Line 137: were these patients suspected of infected heart valves requiring surgical interventions due to infection, or did some of them undergo the procedures for issues other than valve infections?

Lines 155-156: please explain the reason why samples in the negative control groups that tested positive using 16S RC-PCR were excluded from this analysis. What were these contaminating organisms? What did the authors use as a negative control and how was it processed along with real samples? Were these organisms present in all clinical samples too? Were they known ubiquitous environmental contaminant? What exactly was the thought process that resulted in deciding these organisms should be ignored?

Line 159: these additional organisms seemed to be anaerobic flora from mucosal surfaces. While it is true that these organisms could contribute to the infection, chances were those could just be contaminants from the mucosal surfaces. How would the authors decide whether anaerobes identified through 16S RC-PCR all contributed to the infection? How would these results be communicated to the clinicians? Or does it even matter at all because from the clinical condition described in Table 2, for these cases physicians would mostly prescribe antimicrobial agents that cover anaerobes anyway.

Line 235: Sneathia is a gram-negative anaerobe and this did not match the direct gram stain which showed GPC (probably Enterococcus). High suspicion should be considered for this sample when Sneathia was identified. This is something that the authors should discuss. If something is identified through molecular method and does not match what is seen through conventional methods, it should be flagged as a problematic specimen and a consultation should be made with a clinical microbiologist and the clinical team. From lines 257 onward the authors did mention this concern but this reviewer would like to make sure it is clear that these results need to be examined very carefully and it will take a good teamwork between lab and clinicians to make sure patient receives the appropriate treatments.

Lines 261-262: for S24, there was supporting evidence of *S. aureus* so the lab director could consider verbally informing the clinical team of the findings. However, for S45, samples from the surgical site should have been recollected and urogenital samples should have been tested for *N. gonorrhoeae* to rule out disseminated gonococcal infection. Were there follow ups on

these cases?

Line 246: what exactly was used as a negative control?

Line 266: the choice of reference databases is critical to approaches such as this one. Databases chosen should be well maintained and curated. Commercial products may be considered instead of NCBI which is quite disorganized and minimally curated. SILVA is good but has not been updated in a while. Once a database is updated, a mini verification of the new database should be performed to ensure that the new database does not affect the performance of the assay. Another thing to be mindful about is the constant changes in microbial nomenclature. The lab would need to be on top of these things and be aware of updates in the databases used for analysis and to make sure clinicians are aware of these changes.

Lines 281: human DNA depletion followed by deep unbiased metagenomic sequencing may be a more appropriate method if the authors wanted to extend the scope of assay beyond organism identification.

Line 335: The authors did not mention EasySeq, which is a commercial product, until the very end of the manuscript. Should the authors mention this sooner since an established commercial product was used throughout the entire project?

Table 2 item S35: which serology are we talking about?

Table 2 item S52: there really was no clinical info for this case? *Dermacoccus* could be a rare cause of catheter and shunt-associated infections. Did this patient have a VP shunt or any indwelling devices?

Supplemental info:

1. Did the authors see anything within reads that didn't assemble into 16S genes? This reviewer wondered how many human reads there were within the data. What measures did the authors have in place to scrub out human reads before the data are processed through the read classifier?

2. What were the average insert size for PCR products generated by 16S RC-PCR? The authors chose to use the 2x150 kit without tagmentation. Were the authors sure that all PCR products were sequenced in their entirety since the reads were so short? If the PCR product is larger than 300 bp the info in the middle may be missing. Why didn't the authors choose kits that provide longer reads and more data?

Staff Comments:

Preparing Revision Guidelines

Please return the manuscript within 60 days; if you cannot complete the modification within this time period, please contact me. If you do not wish to modify the manuscript and prefer to submit it to another journal, please notify me of your decision immediately so that the manuscript may be formally withdrawn from consideration by Microbiology Spectrum.

Reviewer comments:

Reviewer #1 (Comments for the Author):

In this well written manuscript the authors present a comparison between two methods of 16S rRNA sequencing for identification of bacterial pathogens from clinical specimens. One method is a traditional Sanger sequencing method, and the comparator is an NGS based system utilizing RC-PCR. The authors retrospectively assessed remnant clinical specimens from multiple tissue sources to demonstrate superior performance of the RC-PCR approach. Notably, in instances where more organisms were identified by the RC-PCR approach, the authors performed thorough review of patient cases in a collaborative manner to determine if the identified organisms could be in fact pathogens. This study is limited by the retrospective nature, single center design, and relatively small number of specimens evaluated. However, it is overall well done, and provides interesting insight into a new method for direct-from-specimen identification using 16S sequencing. This is an impressive article and is overall well performed.

That said, I have two major comments about this manuscript:

The first relates to the comparison between the two assays. This is not an apples-to-apples comparison. It is more akin to a comparison between an apple and an extravagantly baked apple pie. Meaning, the intention of the two methods is the same (identify potential pathogens using 16S sequencing) but the methods used are drastically different. All methods used in the RC-PCR approach, from specimen extraction to bioinformatics, are markedly more robust than those used for the Sanger approach. The Sanger approach used one forward and two reverse, somewhat antiquated post-amplification clean-up steps (including gel extraction, which is notorious for loss of amplicons), a simple bioinformatic approach (BLAST), and a problematic database (NCBI nucleotide). In comparison, the RC-PCR approach used six primer pairs, efficient magnetic bead post-PCR clean-up, a robust k-mer based bioinformatic approach, and a curated database (SILVA). Accordingly, it is not surprising that the RC-PCR approach was superior. Could the Sanger approach be improved to similar performance as the RC-PCR approach with optimized primers, sample processing steps, bioinformatic analysis, and use of a more robust database?

We thank the reviewer for the very positive assessment of our study. We fully agree that this is not an apples-to-apples comparison and actually our goal was not to have an equal comparison. In our study, we aimed to demonstrate the clinical utility of the implementation of a more advanced method (RC-PCR) as compared to our current diagnostic test (16S Sanger method) that we use as part of our routine diagnostics (and is widely used in clinical routine microbiology laboratories). Other research groups already assessed the effect of different primer pairs, workflow and bioinformatic analysis within the 16S Sanger method (Baker et al., *Microbiol Methods*, 2003; Yang et al., *BMC Bioinformatics*, 2016; Walker et al., *Scientific reports*, 2020; Kommedal et al., *J Clin Microbiol*, 2009), we aimed here to show the increased potential of a novel method in clinical practice as compared to current practice.

The authors need to address these differences in technique and the potential implications for their results. Additionally, the authors need to clarify why this specific approach to Sanger sequencing was used as a comparator for the RC-PCR method. If this is the Sanger method used in the institutional microbiology laboratory, that needs to be explicitly stated, and it is an

entirely acceptable comparator if so. Otherwise, it seems that a clearly inferior approach was chosen as a comparator, making the study appear somewhat biased from the start.

We compared the RC-PCR method with the Sanger method precisely as used in our current diagnostic laboratory. In the revised manuscript we added “our current diagnostic method using” in the introduction section to explicitly state that Sanger method is our current diagnostic method (lines 112, 222-223).

The second major concern relates to the development and potential marketing of this approach. No conflicts of interest are declared by any authors, yet lines 334-336 state, "The 16S RC-PCR method as developed in this study has recently been released as EasySeq™ 16S rRNA Bacterial ID kit by NimaGen B.V., Nijmegen, The Netherlands". Accordingly, clarity is needed as to how the authors came to study this approach if it was already in development for commercial offering. Do the authors have relationships with NimaGen? Did the authors have access to this product before released as a commercial product and was evaluating performance prior to release? Or, did the authors develop this product that was then purchased by NimaGen for release? These questions need to be clarified to ensure there are indeed no conflicts of interest.

Authors of this paper co-developed, designed, and optimized the assay together with NimaGen B.V. from an academic perspective. This was before release of the EasySeq product. Furthermore, authors designed the study and developed the bioinformatics independently from NimaGen B.V. Results were evaluated and written down without interference of NimaGen B.V. This has been added to the revised version, Supplementary materials lines 40-43, 51-52.

Both the authors of this manuscript and the Radboudumc have no financial benefits what so ever from this collaboration. Our collaboration with NimaGen B.V. had as sole purpose to improve our current routine diagnostics.

Additional minor comments are as follows:

There is an overall lack of quality control data to demonstrate that the 16S amplicons generated by upstream processes are of similar quality before undergoing Sanger or Illumina based sequencing. Please include information in the methods regarding how the efficiency of gel extraction was assessed in the Sanger method prior to Sanger sequencing. How do you know you had enough amplicon remaining after the extraction steps to even sequence anything?

In the Sanger method, after clean-up, the DNA is once more put on gel for quality control. Before performing Sanger sequencing, DNA concentrations are measured using the Qubit. Dependent on the product size, a specified concentration of DNA is used for Sanger sequencing. We added this information to the Methods section (Supplementary information, lines 22-25).

It is impossible to have similar quality amplicons before sequencing because the first step of amplicon construction between PCR and RC-PCR is already fundamentally different. As indicated above, our aim was to compare the currently used method in routine diagnostics of 16S Sanger to a different method that could be implemented in clinical practice rather than performing an apples to apples comparison. We therefore agree with the reviewer the importance of more explicitly stating that the 16S Sanger method is the method we currently use in routine diagnostics (line 112, 222-223).

The sentence spanning lines 83-86 needs to be revised for accuracy, addressing the following

points: 1) use of the phrase "often fails" requires a reference to define how often this fails, and 2) "most-commonly used NGS platforms" differs by laboratory, so this should be clarified to describe short-read vs long-read sequencing approaches.

1) Based on our own experience amplification of large fragments rarely succeeds in clinical samples when bacterial loads are low, in contrast to isolates when bacterial loads are high. We rephrased the sentence by removing 'often' (line 85).

2) We agree with the reviewer and replaced "most-commonly used NGS platforms" by 'short-read sequencing technologies' (line 86).

The sentence spanning lines 88-90 needs to be revised for accuracy. A reference is needed to define how "most microbiology laboratories" perform 16S sequencing. Does this statement apply globally, only to European or US labs, or some combination thereof? Additionally, this statements grossly oversimplifies the complex methods that comprise 16S sequencing. It suggests "most microbiology laboratories" perform similar methods, which is incorrect. There is not an accepted standard practice for this identification approach. As such different laboratories use different primers, sequencing methodologies, bioinformatic strategies, and databases.

We agree with the reviewer, there is no international standard for the use of 16S primers for clinical applications and as can be understood from the literature there is a great variety between different laboratories in the 16S protocol used. We revised the sentence accordingly (lines 89-93). We replaced, "Currently" by "Since several Decades", removed "most" and added two references including a review about 16S Sanger for the detection of pathogens in clinical samples.

What is meant by "a single closed tube system" on line 103? This seems to contrast the 96 well plate that is presented in Figure 1.

Changed "a single closed tube system" to "closed-tube system available as 96-well plate format" (line 104).

Line 137: please change "send" to "sent"

This has been corrected.

Additional information is needed regarding the clinical specimens utilized in this specimen (both the heart valves and the additional specimens). Specifically, where these samples handled with strictly sterile methods prior to evaluation? This is very important to describe given the potential for detection of contaminants that could be misidentified as potential pathogens (separate from those detected by use of appropriate negative controls).

Specimen collection is performed during treatment or surgery from the patient according to Standard Operating procedures. These protocols enhance sterile sampling as much as possible. We added this information to the Materials and Methods section (lines 335-337). However, contamination cannot be fully avoided. The results reflected in the manuscript are assessed in the same manner between 16S Sanger and 16S RC-PCR as both methods were applied on the same specimen. Furthermore, contaminants observed are introduced by reagents and chemicals used, as described by Salter et al., BMC Biology, 2014.

We therefore feel that it is strictly necessary to include continues negative controls with each test, as mentioned in the Discussion section (line 266-268). In addition, this also points out the need for multidisciplinary consultation before confirming the results (line 278-282).

Please elaborate on the professions of those involved in the "multidisciplinary consultation" described on line 173.

Line 193: Added "comprised of a medical microbiologist, molecular expert, and a bioinformatician."

Lines 202-203 need to be revised for accuracy for the same reasons mentioned above. Stating the "often used 16S Sanger approach" assumes the same method is used by laboratories performing such testing, but that is not accurate. Laboratories use techniques that differ in all steps of the process and as such cannot be reduced with such simplicity.

We agree with the reviewer, and removed "the currently often" and replaced with "our currently" (lines 222-223). To really state that this comparison of results reflects our specific used 16S Sanger method.

Although Panels A and B in Figure 2 are visually pleasing representations of the K-mer analysis (using what seems to be Pavian, or a similar program), more exposition is needed in the body text or the figure legend to explain the benefits of taxonomic identification using this bioinformatic approach. As it stands, it may be difficult for a reader without bioinformatic experience to appreciate the additional information provided by such an approach as compared to simply utilizing Blast and the NCBI nucleotide database (as we performed for the Sanger approach).

For clarification we added "The added value of the automated taxonomic information retrieved from the RC-PCR Classifier is that it uses the curated SILVA database and taxonomy. Presenting the results this way makes interpretation more straightforward compared to NCBI BLAST performed using 16S Sanger." to the caption of Figure 2.

Please clarify in the figure legend what is meant by "Abundance" in the heatmaps of Figure 2. Does this represent the relative abundance (percentage) of the given organism in the specimen as compared to all reads, all microbial reads, or something else?

Line 384-385: Added to caption

"Abundance: Is calculated as the relative percentage fragments of a given hit compared to all quality filtered fragments."

In figure 2 D, do the colored bars next to "Identified species" (orange), "Background" (grey), "Culture positive" (yellow), and "Culture-negative" (green) correspond to data in the figure? If these colored bards to not convey information, then please remove them.

Colored bars are removed in figure 2D.

Reviewer #2 (Comments for the Author):

General comments: This manuscript by Moorlag et al. explored the use of 16S rRNA reverse complement polymerase chain reaction (16S RC-PCR) which was a commercially available reagents/method for the identification of bacterial species. The manuscript demonstrated that

their approach using EasySeq was more successful in identification of bacteria from culture-negative clinical specimens. The word "reverse" kind of confused this reviewer a bit because this reviewer immediately thought of reverse transcription. There were studies that if extracted nucleic acid went through reverse transcription first before 16S PCR, the sensitivity could be higher since 16S rRNA genes are typically multi copy and living bacterial cells make a lot of rRNA as they grow and divide, and this was what this reviewer thought this study was about. Once this reviewer finally understood the technique, this approach was quite smart because it enabled specificity of amplification to useful variable segments of 16S rRNA gene while including the indexing and adapter attachment at the same time. Although CLIA does not apply to the Netherlands, the authors should clearly demonstrate the performance of the assay, including reproducibility (which this reviewer believed the authors had done some of it), limit of detection (CFU/mL), interfering substances, stability, etc. This reviewer was also curious as to how different types of samples were processed. For example, a heart valve or tissue may be eluted and/or homogenized in sterile saline for culture. Was the exact same processed materials used for sequencing and was it done immediately in parallel (culture, Sanger, 16S RC-PCR)?

16S Sanger and 16S RC PCR were done in parallel for the analysis on heart valves presented in Table 1 and were performed on the exact same processed materials. For the clinical specimens presented in Table 2 and 3 first culture and Sanger were performed and afterwards 16S RC-PCR, on the same specimen. To demonstrate reproducibility all samples were performed in triplo for 16S RC PCR and demonstrated similar results.

For this rebuttal additional experiments were conducted to measure the Limit of detection (LOD). ZymoBIOMICS™ Microbial Community Standard II (Log Distribution) and ZymoBIOMICS Microbial Community DNA Standard II (Log Distribution) were tested in triplicate. The ZymoBIOMICS products used are designed to measure the LOD in 16S abundance in a polymicrobial standard. We conducted extra dilution series to add extra complexity to the experiment and be able to get more insight into the number of cells in the LOD under a polymicrobial standard that can be detected. We added the results of these experiments to the revised manuscript (lines 140-151, Supplementary Figure S2 and Table S2).

The authors mentioned that sample was processed according to their SOPs but this reviewer was just curious about the workflow. Other comments are below.

For the extraction of DNA from tissue samples a small piece in saline was added to a tube containing MagNA Lyser Green Beads (Roche) (20s 6500x). A volume of 200 µL was used as input for isolation. Lysis was performed using the MagNA Pure 96 DNA and Viral NA Small Volume Kit (Roche) (see Supplementary information lines 8-12)).

Comments

Line 60-61: It should be clarified that increased number of bacterial infections won't necessarily result in better treatment outcomes. It depends on whether the assay truly identifies whether an infection is the cause of the illness and if so, which organism(s) are the etiologic agent(s).

Line 61: Removed "thereby" and replaced it with "this in combination with adequate treatment could" to show it won't necessarily result in better treatment outcomes.

Lines 84-85: It was unclear to this reviewer why this could not be feasible with most commonly used NGS platforms. Please clarify why and which platforms the authors were talking about.

Here we mean short-read platforms. We made this more clear in the revised version, without picking a single platform because all short-read platforms suffer of the same limitations.

Lines 85-86: Changed, “bacteria and is not feasible on the most commonly used NGS-platforms” to “bacteria and sequencing is not feasible using short-read sequencing technologies”

Line 86: which platform is that?

Made it more specific without picking a single platform because all short-read platforms suffer of the same limitations. Line 86-87: Changed, “Analyses of shorter fragments,” to “Analyses and sequencing of short-reads”

Line 124: please explain what they authors meant by "in vivo"

Line 126, 128: Authors removed “in vivo” because we agree with the reviewer that it is a confusing term in this sentence and changed it to “laboratory derived”. Furthermore, added “made using clinical isolates” to better specify the content of the mock community. Later in the same paragraph, , authors added “laboratory derived” when discussing the results of that lab derived mock and added “commercial” to the other mock which is commercially available for clarification.

Line 137: were these patients suspected of infected heart valves requiring surgical interventions due to infection, or did some of them undergo the procedures for issues other than valve infections?

These patients were suspected of infected heart valves (endocarditis). We clarified this in the text (line 155).

Lines 155-156: please explain the reason why samples in the negative control groups that tested positive using 16S RC-PCR were excluded from this analysis. What were these contaminating organisms? What did the authors use as a negative control and how was it processed along with real samples? Were these organisms present in all clinical samples too? Were they known ubiquitous environmental contaminant? What exactly was the thought process that resulted in deciding these organisms should be ignored?

Most contaminants observed in the study are introduced by reagents and chemicals used. This phenomenon is a known downside of NGS-based detection and multiple studies have shown the presence of DNA in reagents used during the process from DNA isolation until sequencing (Salter et al., BMC Biology, 2014; Glassing et al., Gut Pathog, 2016).

It is therefore not uncommon to include negative control samples to be able to subtract these findings from the findings in the clinical samples. These negative controls are samples that have been processed simultaneously with the clinical samples and processed using the full protocol from DNA isolation to RC-PCR but without adding material. An overview of the species detected in the negative controls is presented in Figure S1. Most species are environmental contaminants not clearly associated with human infection and can be regarded as contamination. However, as mentioned in the Discussion section (line 272-275) four clinical samples contained species that can be clinically significant microorganisms but were also found in at least one negative control, for example Cutibacterium sp. As discussed in lines 275-277, abundance and the use of internal controls together with the clinical context can be helpful to ensure correct interpretation. As we aimed in our paper to compare clinical relevant findings as is most important in clinical practice,

rather than the detection of any species by RC-PCR vs Sanger, we decided beforehand to exclude samples with microorganisms that were also found in at least in one negative control, to avoid an overrepresentation of our findings in this study.

Line 159: these additional organisms seemed to be anaerobic flora from mucosal surfaces. While it is true that these organisms could contribute to the infection, chances were those could just be contaminants from the mucosal surfaces. How would the authors decide whether anaerobes identified through 16S RC-PCR all contributed to the infection? How would these results be communicated to the clinicians? Or does it even matter at all because from the clinical condition described in Table 2, for these cases physicians would mostly prescribe antimicrobial agents that cover anaerobes anyway.

Authors are aware of difficulties interpreting results that could contribute to the infection. This is not only an issue in those cases where molecular techniques are used, also when culture is used as diagnostic method the cultured bacteria might be contributing to the infection, or be colonization of the host. Therefore, the clinical context, response to current treatment etcetera have to be taken into account and we strongly advocate that all results have to be discussed using a multidisciplinary consultation comprised of a medical microbiologist, molecular expert, and a bioinformatician before communicating to a clinician. For the specific clinical cases in Table 2, we would discuss the presence of anaerobes with the clinician as in these potentially life-threatening conditions anaerobes might be contributors to the infection. It depends on the condition of the patient, the response to current therapy etc. how to interpret the results for the specific patient.

Line 235: Sneathia is a gram-negative anaerobe and this did not match the direct gram stain which showed GPC (probably Enterococcus). High suspicion should be considered for this sample when Sneathia was identified. This is something that the authors should discuss. If something is identified through molecular method and does not match what is seen through conventional methods, it should be flagged as a problematic specimen and a consultation should be made with a clinical microbiologist and the clinical team. From lines 257 onward the authors did mention this concern but this reviewer would like to make sure it is clear that these results need to be examined very carefully and it will take a good teamwork between lab and clinicians to make sure patient receives the appropriate treatments.

We cannot agree more with the reviewer that a very careful examination (by a medical microbiologist, bioinformatician, molecular expert) of results is crucial, followed by a consultation between the clinical microbiologist and the clinical team. In the revised version of the manuscript we made this more clear (lines 279-282).

As for the Sneathia case, these steps were carefully performed at that time. The multidisciplinary team (involving also the clinician) decided after multiple consultations to start treatment directed at Sneathia sp as the condition of the patient was deteriorating, was not responding to current therapy and literature provided evidence for a potential role of Sneathia sp in the condition of the patient.

Lines 261-262: for S24, there was supporting evidence of *S. aureus* so the lab director could consider verbally informing the clinical team of the findings. However, for S45, samples from the surgical site should have been recollected and urogenital samples should have been tested for

N. gonorrhoeae to rule out disseminated gonococcal infection. Were there follow ups on these cases?

We agree with the reviewer that these results could be considered to communicate to the clinical team and added this to the revised manuscript (lines 286-287). Regarding S24, patient was treated for S. aureus endocarditis. Regarding S45, when RC-PCR was performed on the material the patient already recovered so no follow-up diagnostics were performed unfortunately targeted at N. gonorrhoeae.

Line 246: what exactly was used as a negative control?

These negative controls are samples that have been processed simultaneously with the clinical samples and processed using the full protocol from DNA isolation to RC-PCR but without adding material. This information has been added to the caption in Figure S1 (Supplementary information).

Line 266: the choice of reference databases is critical to approaches such as this one. Databases chosen should be well maintained and curated. Commercial products may be considered instead of NCBI which is quite disorganized and minimally curated. SILVA is good but has not been updated in a while. Once a database is updated, a mini verification of the new database should be performed to ensure that the new database does not affect the performance of the assay. Another thing to be mindful about is the constant changes in microbial nomenclature. The lab would need to be on top of these things and be aware of updates in the databases used for analysis and to make sure clinicians are aware of these changes.

The authors thank the reviewer for his thoughts on the reference database subject. Authors also agree on this topic and internally always verify and validate if a database is updated. This is standard procedure for our bioinformatics team and is always checked by our QC officer.

Lines 281: human DNA depletion followed by deep unbiased metagenomic sequencing may be a more appropriate method if the authors wanted to extend the scope of assay beyond organism identification.

Line 304-306: Added the line "Another appropriate but more expensive and less sensitive method would be to perform Human DNA depletion followed by metagenome sequencing referred to as clinical metagenomics (Chiu and Miller, Nature Reviews Genetics, 2019).

Line 335: The authors did not mention EasySeq, which is a commercial product, until the very end of the manuscript. Should the authors mention this sooner since an established commercial product was used throughout the entire project?

Authors of this paper co-developed, designed, and optimized the assay together with NimaGen B.V before release of the EasySeq product. Furthermore, authors designed the study and developed the bioinformatics independently from NimaGen B.V. Outcome of the assay was evaluated and written down without interference of NimaGen B.V. This has been added to the revised version, Supplementary materials lines 40-43, 51-52.

We engaged this collaboration because we wanted to innovate our diagnostics and the RC technology.

Table 2 item S35: which serology are we talking about?

Anti-DNase B and ASO antibodies. We added this to Table 3.

Table 2 item S52: there really was no clinical info for this case? Dermacoccus could be a rare cause of catheter and shunt-associated infections. Did this patient have a VP shunt or any indwelling devices?

This case involves a patient from a different hospital from which we received material to perform 16S. For this reason, we unfortunately do not have any clinical information as the 16S Sanger result at that time remained negative so there was no consultation.

Supplemental info:

1. Did the authors see anything within reads that didn't assemble into 16S genes? This reviewer wondered how many human reads there were within the data. What measures did the authors have in place to scrub out human reads before the data are processed through the read classifier?

The bioinformatic analysis uses a read cleaning tool that removes low quality sequence reads. Almost no human reads have been sequenced and the method used for classifying the reads to the SILVA database selects the 16S targeting reads with certain quality so scrubbing out the human reads is redundant.

2. What were the average insert size for PCR products generated by 16S RC-PCR? The authors chose the use the 2x150 kit without tagmentation. Were the authors sure that all PCR products were sequenced in their entirety since the reads were so short? If the PCR product is larger than 300 bp the info in the middle may be missing. Why didn't the authors choose kits that provide longer reads and more data?

The reviewer is correct that we used 2x150. We used an Illumina MiniSeq. 1) for its faster turnaround time and 2) because this is also the workflow that we use for our SARS-CoV-2 mutation screening (Coolen et al., J Clin Virol, 2021). Using larger sequencing fragments would indeed cover the middle better and is therefore an approach that can be used but is not tested by us because of previous mentioned reasons.

April 28, 2023

Dr. Jordy P. M. Coolen
Radboudumc
Medical Microbiology
Geert Grooteplein Zuid 10
Nijmegen 6525 GA
Netherlands

Re: Spectrum04483-22R1 (Targeting 16S rRNA by reverse complement PCR-next-generation-sequencing: specific and sensitive detection and identification of microbes directly in clinical samples)

Dear Dr. Jordy P. M. Coolen:

Thank you for submitting your manuscript to Microbiology Spectrum. As you will see your paper is very close to acceptance. Following review, I believe that you adequately addressed Reviewer 1's concern regarding Conflict of Interest. However, as these statements were added to the supplemental material rather than the primary manuscript, I am asking that the sections on page 14 of the manuscript be updated to reflect these new statements.

Please modify the manuscript along the lines I have recommended. As these revisions are quite minor, I expect that you should be able to turn in the revised paper in less than 30 days, if not sooner. If your manuscript was reviewed, you will find the reviewers' comments below.

When submitting the revised version of your paper, please provide (1) point-by-point responses to the issues raised by the reviewers as file type "Response to Reviewers," not in your cover letter, and (2) a PDF file that indicates the changes from the original submission (by highlighting or underlining the changes) as file type "Marked Up Manuscript - For Review Only". Please use this link to submit your revised manuscript. Detailed instructions on submitting your revised paper are below.

Link Not Available

Sincerely,

Paul Luethy

Reviewer comments:

Preparing Revision Guidelines

- Point-by-point responses to the issues raised by the reviewers in a file named "Response to Reviewers," NOT IN YOUR COVER LETTER.
- Upload a compare copy of the manuscript (without figures) as a "Marked-Up Manuscript" file.

- Each figure must be uploaded as a separate file, and any multipanel figures must be assembled into one file.
- Manuscript: A .DOC version of the revised manuscript
- Figures: Editable, high-resolution, individual figure files are required at revision, TIFF or EPS files are preferred

Please return the manuscript within 60 days; if you cannot complete the modification within this time period, please contact me. If you do not wish to modify the manuscript and prefer to submit it to another journal, please notify me of your decision immediately so that the manuscript may be formally withdrawn from consideration by Microbiology Spectrum.

Reviewer comments:

Reviewer #1 (Comments for the Author):

In this well written manuscript the authors present a comparison between two methods of 16S rRNA sequencing for identification of bacterial pathogens from clinical specimens. One method is a traditional Sanger sequencing method, and the comparator is an NGS based system utilizing RC-PCR. The authors retrospectively assessed remnant clinical specimens from multiple tissue sources to demonstrate superior performance of the RC-PCR approach. Notably, in instances where more organisms were identified by the RC-PCR approach, the authors performed thorough review of patient cases in a collaborative manner to determine if the identified organisms could be in fact pathogens. This study is limited by the retrospective nature, single center design, and relatively small number of specimens evaluated. However, it is overall well done, and provides interesting insight into a new method for direct-from-specimen identification using 16S sequencing. This is an impressive article and is overall well performed.

That said, I have two major comments about this manuscript:

The first relates to the comparison between the two assays. This is not an apples-to-apples comparison. It is more akin to a comparison between an apple and an extravagantly baked apple pie. Meaning, the intention of the two methods is the same (identify potential pathogens using 16S sequencing) but the methods used are drastically different. All methods used in the RC-PCR approach, from specimen extraction to bioinformatics, are markedly more robust than those used for the Sanger approach. The Sanger approach used one forward and two reverse, somewhat antiquated post-amplification clean-up steps (including gel extraction, which is notorious for loss of amplicons), a simple bioinformatic approach (BLAST), and a problematic database (NCBI nucleotide). In comparison, the RC-PCR approach used six primer pairs, efficient magnetic bead post-PCR clean-up, a robust k-mer based bioinformatic approach, and a curated database (SILVA). Accordingly, it is not surprising that the RC-PCR approach was superior. Could the Sanger approach be improved to similar performance as the RC-PCR approach with optimized primers, sample processing steps, bioinformatic analysis, and use of a more robust database?

We thank the reviewer for the very positive assessment of our study. We fully agree that this is not an apples-to-apples comparison and actually our goal was not to have an equal comparison. In our study, we aimed to demonstrate the clinical utility of the implementation of a more advanced method (RC-PCR) as compared to our current diagnostic test (16S Sanger method) that we use as part of our routine diagnostics (and is widely used in clinical routine microbiology laboratories). Other research groups already assessed the effect of different primer pairs, workflow and bioinformatic analysis within the 16S Sanger method (Baker et al., *Microbiol Methods*, 2003; Yang et al., *BMC Bioinformatics*, 2016; Walker et al., *Scientific reports*, 2020; Kommedal et al., *J Clin Microbiol*, 2009), we aimed here to show the increased potential of a novel method in clinical practice as compared to current practice.

The authors need to address these differences in technique and the potential implications for their results. Additionally, the authors need to clarify why this specific approach to Sanger sequencing was used as a comparator for the RC-PCR method. If this is the Sanger method used in the institutional microbiology laboratory, that needs to be explicitly stated, and it is an

entirely acceptable comparator if so. Otherwise, it seems that a clearly inferior approach was chosen as a comparator, making the study appear somewhat biased from the start.

We compared the RC-PCR method with the Sanger method precisely as used in our current diagnostic laboratory. In the revised manuscript we added “our current diagnostic method using” in the introduction section to explicitly state that Sanger method is our current diagnostic method (lines 112, 222-223).

The second major concern relates to the development and potential marketing of this approach. No conflicts of interest are declared by any authors, yet lines 334-336 state, "The 16S RC-PCR method as developed in this study has recently been released as EasySeq™ 16S rRNA Bacterial ID kit by NimaGen B.V., Nijmegen, The Netherlands". Accordingly, clarity is needed as to how the authors came to study this approach if it was already in development for commercial offering. Do the authors have relationships with NimaGen? Did the authors have access to this product before released as a commercial product and was evaluating performance prior to release? Or, did the authors develop this product that was then purchased by NimaGen for release? These questions need to be clarified to ensure there are indeed no conflicts of interest.

Authors of this paper co-developed, designed, and optimized the assay together with NimaGen B.V. from an academic perspective. This was before release of the EasySeq product. Furthermore, authors designed the study and developed the bioinformatics independently from NimaGen B.V. Results were evaluated and written down without interference of NimaGen B.V. This has been added to the revised version, page 14.

Both the authors of this manuscript and the Radboudumc have no financial benefits what so ever from this collaboration. Our collaboration with NimaGen B.V. had as sole purpose to improve our current routine diagnostics.

Additional minor comments are as follows:

There is an overall lack of quality control data to demonstrate that the 16S amplicons generated by upstream processes are of similar quality before undergoing Sanger or Illumina based sequencing. Please include information in the methods regarding how the efficiency of gel extraction was assessed in the Sanger method prior to Sanger sequencing. How do you know you had enough amplicon remaining after the extraction steps to even sequence anything?

In the Sanger method, after clean-up, the DNA is once more put on gel for quality control. Before performing Sanger sequencing, DNA concentrations are measured using the Qubit. Dependent on the product size, a specified concentration of DNA is used for Sanger sequencing. We added this information to the Methods section (Supplementary information, lines 22-25).

It is impossible to have similar quality amplicons before sequencing because the first step of amplicon construction between PCR and RC-PCR is already fundamentally different. As indicated above, our aim was to compare the currently used method in routine diagnostics of 16S Sanger to a different method that could be implemented in clinical practice rather than performing an apples to apples comparison. We therefore agree with the reviewer the importance of more explicitly stating that the 16S Sanger method is the method we currently use in routine diagnostics (line 112, 222-223).

The sentence spanning lines 83-86 needs to be revised for accuracy, addressing the following

points: 1) use of the phrase "often fails" requires a reference to define how often this fails, and 2) "most-commonly used NGS platforms" differs by laboratory, so this should be clarified to describe short-read vs long-read sequencing approaches.

1) Based on our own experience amplification of large fragments rarely succeeds in clinical samples when bacterial loads are low, in contrast to isolates when bacterial loads are high. We rephrased the sentence by removing 'often' (line 85).

2) We agree with the reviewer and replaced "most-commonly used NGS platforms" by 'short-read sequencing technologies' (line 86).

The sentence spanning lines 88-90 needs to be revised for accuracy. A reference is needed to define how "most microbiology laboratories" perform 16S sequencing. Does this statement apply globally, only to European or US labs, or some combination thereof? Additionally, this statements grossly oversimplifies the complex methods that comprise 16S sequencing. It suggests "most microbiology laboratories" perform similar methods, which is incorrect. There is not an accepted standard practice for this identification approach. As such different laboratories use different primers, sequencing methodologies, bioinformatic strategies, and databases.

We agree with the reviewer, there is no international standard for the use of 16S primers for clinical applications and as can be understood from the literature there is a great variety between different laboratories in the 16S protocol used. We revised the sentence accordingly (lines 89-93). We replaced, "Currently" by "Since several Decades", removed "most" and added two references including a review about 16S Sanger for the detection of pathogens in clinical samples.

What is meant by "a single closed tube system" on line 103? This seems to contrast the 96 well plate that is presented in Figure 1.

Changed "a single closed tube system" to "closed-tube system available as 96-well plate format" (line 104).

Line 137: please change "send" to "sent"

This has been corrected.

Additional information is needed regarding the clinical specimens utilized in this specimen (both the heart valves and the additional specimens). Specifically, where these samples handled with strictly sterile methods prior to evaluation? This is very important to describe given the potential for detection of contaminants that could be misidentified as potential pathogens (separate from those detected by use of appropriate negative controls).

Specimen collection is performed during treatment or surgery from the patient according to Standard Operating procedures. These protocols enhance sterile sampling as much as possible. We added this information to the Materials and Methods section (lines 335-337). However, contamination cannot be fully avoided. The results reflected in the manuscript are assessed in the same manner between 16S Sanger and 16S RC-PCR as both methods were applied on the same specimen. Furthermore, contaminants observed are introduced by reagents and chemicals used, as described by Salter et al., BMC Biology, 2014.

We therefore feel that it is strictly necessary to include continues negative controls with each test, as mentioned in the Discussion section (line 266-268). In addition, this also points out the need for multidisciplinary consultation before confirming the results (line 278-282).

Please elaborate on the professions of those involved in the "multidisciplinary consultation" described on line 173.

Line 193: Added "comprised of a medical microbiologist, molecular expert, and a bioinformatician."

Lines 202-203 need to be revised for accuracy for the same reasons mentioned above. Stating the "often used 16S Sanger approach" assumes the same method is used by laboratories performing such testing, but that is not accurate. Laboratories use techniques that differ in all steps of the process and as such cannot be reduced with such simplicity.

We agree with the reviewer, and removed "the currently often" and replaced with "our currently" (lines 222-223). To really state that this comparison of results reflects our specific used 16S Sanger method.

Although Panels A and B in Figure 2 are visually pleasing representations of the K-mer analysis (using what seems to be Pavian, or a similar program), more exposition is needed in the body text or the figure legend to explain the benefits of taxonomic identification using this bioinformatic approach. As it stands, it may be difficult for a reader without bioinformatic experience to appreciate the additional information provided by such an approach as compared to simply utilizing Blast and the NCBI nucleotide database (as we performed for the Sanger approach).

For clarification we added "The added value of the automated taxonomic information retrieved from the RC-PCR Classifier is that it uses the curated SILVA database and taxonomy. Presenting the results this way makes interpretation more straightforward compared to NCBI BLAST performed using 16S Sanger." to the caption of Figure 2.

Please clarify in the figure legend what is meant by "Abundance" in the heatmaps of Figure 2. Does this represent the relative abundance (percentage) of the given organism in the specimen as compared to all reads, all microbial reads, or something else?

Line 384-385: Added to caption

"Abundance: Is calculated as the relative percentage fragments of a given hit compared to all quality filtered fragments."

In figure 2 D, do the colored bars next to "Identified species" (orange), "Background" (grey), "Culture positive" (yellow), and "Culture-negative" (green) correspond to data in the figure? If these colored bards to not convey information, then please remove them.

Colored bars are removed in figure 2D.

Reviewer #2 (Comments for the Author):

General comments: This manuscript by Moorlag et al. explored the use of 16S rRNA reverse complement polymerase chain reaction (16S RC-PCR) which was a commercially available reagents/method for the identification of bacterial species. The manuscript demonstrated that

their approach using EasySeq was more successful in identification of bacteria from culture-negative clinical specimens. The word "reverse" kind of confused this reviewer a bit because this reviewer immediately thought of reverse transcription. There were studies that if extracted nucleic acid went through reverse transcription first before 16S PCR, the sensitivity could be higher since 16S rRNA genes are typically multi copy and living bacterial cells make a lot of rRNA as they grow and divide, and this was what this reviewer thought this study was about. Once this reviewer finally understood the technique, this approach was quite smart because it enabled specificity of amplification to useful variable segments of 16S rRNA gene while including the indexing and adapter attachment at the same time. Although CLIA does not apply to the Netherlands, the authors should clearly demonstrate the performance of the assay, including reproducibility (which this reviewer believed the authors had done some of it), limit of detection (CFU/mL), interfering substances, stability, etc. This reviewer was also curious as to how different types of samples were processed. For example, a heart valve or tissue may be eluted and/or homogenized in sterile saline for culture. Was the exact same processed materials used for sequencing and was it done immediately in parallel (culture, Sanger, 16S RC-PCR)?

16S Sanger and 16S RC PCR were done in parallel for the analysis on heart valves presented in Table 1 and were performed on the exact same processed materials. For the clinical specimens presented in Table 2 and 3 first culture and Sanger were performed and afterwards 16S RC-PCR, on the same specimen. To demonstrate reproducibility all samples were performed in triplo for 16S RC PCR and demonstrated similar results.

For this rebuttal additional experiments were conducted to measure the Limit of detection (LOD). ZymoBIOMICS™ Microbial Community Standard II (Log Distribution) and ZymoBIOMICS Microbial Community DNA Standard II (Log Distribution) were tested in triplicate. The ZymoBIOMICS products used are designed to measure the LOD in 16S abundance in a polymicrobial standard. We conducted extra dilution series to add extra complexity to the experiment and be able to get more insight into the number of cells in the LOD under a polymicrobial standard that can be detected. We added the results of these experiments to the revised manuscript (lines 140-151, Supplementary Figure S2 and Table S2).

The authors mentioned that sample was processed according to their SOPs but this reviewer was just curious about the workflow. Other comments are below.

For the extraction of DNA from tissue samples a small piece in saline was added to a tube containing MagNA Lyser Green Beads (Roche) (20s 6500x). A volume of 200 µL was used as input for isolation. Lysis was performed using the MagNA Pure 96 DNA and Viral NA Small Volume Kit (Roche) (see Supplementary information lines 8-12)).

Comments

Line 60-61: It should be clarified that increased number of bacterial infections won't necessarily result in better treatment outcomes. It depends on whether the assay truly identifies whether an infection is the cause of the illness and if so, which organism(s) are the etiologic agent(s).

Line 61: Removed "thereby" and replaced it with "this in combination with adequate treatment could" to show it won't necessarily result in better treatment outcomes.

Lines 84-85: It was unclear to this reviewer why this could not be feasible with most commonly used NGS platforms. Please clarify why and which platforms the authors were talking about.

Here we mean short-read platforms. We made this more clear in the revised version, without picking a single platform because all short-read platforms suffer of the same limitations.

Lines 85-86: Changed, “bacteria and is not feasible on the most commonly used NGS-platforms” to “bacteria and sequencing is not feasible using short-read sequencing technologies”

Line 86: which platform is that?

Made it more specific without picking a single platform because all short-read platforms suffer of the same limitations. Line 86-87: Changed, “Analyses of shorter fragments,” to “Analyses and sequencing of short-reads”

Line 124: please explain what they authors meant by "in vivo"

Line 126, 128: Authors removed “in vivo” because we agree with the reviewer that it is a confusing term in this sentence and changed it to “laboratory derived”. Furthermore, added “made using clinical isolates” to better specify the content of the mock community. Later in the same paragraph, , authors added “laboratory derived” when discussing the results of that lab derived mock and added “commercial” to the other mock which is commercially available for clarification.

Line 137: were these patients suspected of infected heart valves requiring surgical interventions due to infection, or did some of them undergo the procedures for issues other than valve infections?

These patients were suspected of infected heart valves (endocarditis). We clarified this in the text (line 155).

Lines 155-156: please explain the reason why samples in the negative control groups that tested positive using 16S RC-PCR were excluded from this analysis. What were these contaminating organisms? What did the authors use as a negative control and how was it processed along with real samples? Were these organisms present in all clinical samples too? Were they known ubiquitous environmental contaminant? What exactly was the thought process that resulted in deciding these organisms should be ignored?

Most contaminants observed in the study are introduced by reagents and chemicals used. This phenomenon is a known downside of NGS-based detection and multiple studies have shown the presence of DNA in reagents used during the process from DNA isolation until sequencing (Salter et al., BMC Biology, 2014; Glassing et al., Gut Pathog, 2016).

It is therefore not uncommon to include negative control samples to be able to subtract these findings from the findings in the clinical samples. These negative controls are samples that have been processed simultaneously with the clinical samples and processed using the full protocol from DNA isolation to RC-PCR but without adding material. An overview of the species detected in the negative controls is presented in Figure S1. Most species are environmental contaminants not clearly associated with human infection and can be regarded as contamination. However, as mentioned in the Discussion section (line 272-275) four clinical samples contained species that can be clinically significant microorganisms but were also found in at least one negative control, for example Cutibacterium sp. As discussed in lines 275-277, abundance and the use of internal controls together with the clinical context can be helpful to ensure correct interpretation. As we aimed in our paper to compare clinical relevant findings as is most important in clinical practice,

rather than the detection of any species by RC-PCR vs Sanger, we decided beforehand to exclude samples with microorganisms that were also found in at least in one negative control, to avoid an overrepresentation of our findings in this study.

Line 159: these additional organisms seemed to be anaerobic flora from mucosal surfaces. While it is true that these organisms could contribute to the infection, chances were those could just be contaminants from the mucosal surfaces. How would the authors decide whether anaerobes identified through 16S RC-PCR all contributed to the infection? How would these results be communicated to the clinicians? Or does it even matter at all because from the clinical condition described in Table 2, for these cases physicians would mostly prescribe antimicrobial agents that cover anaerobes anyway.

Authors are aware of difficulties interpreting results that could contribute to the infection. This is not only an issue in those cases where molecular techniques are used, also when culture is used as diagnostic method the cultured bacteria might be contributing to the infection, or be colonization of the host. Therefore, the clinical context, response to current treatment etcetera have to be taken into account and we strongly advocate that all results have to be discussed using a multidisciplinary consultation comprised of a medical microbiologist, molecular expert, and a bioinformatician before communicating to a clinician. For the specific clinical cases in Table 2, we would discuss the presence of anaerobes with the clinician as in these potentially life-threatening conditions anaerobes might be contributors to the infection. It depends on the condition of the patient, the response to current therapy etc. how to interpret the results for the specific patient.

Line 235: Sneathia is a gram-negative anaerobe and this did not match the direct gram stain which showed GPC (probably Enterococcus). High suspicion should be considered for this sample when Sneathia was identified. This is something that the authors should discuss. If something is identified through molecular method and does not match what is seen through conventional methods, it should be flagged as a problematic specimen and a consultation should be made with a clinical microbiologist and the clinical team. From lines 257 onward the authors did mention this concern but this reviewer would like to make sure it is clear that these results need to be examined very carefully and it will take a good teamwork between lab and clinicians to make sure patient receives the appropriate treatments.

We cannot agree more with the reviewer that a very careful examination (by a medical microbiologist, bioinformatician, molecular expert) of results is crucial, followed by a consultation between the clinical microbiologist and the clinical team. In the revised version of the manuscript we made this more clear (lines 279-282).

As for the Sneathia case, these steps were carefully performed at that time. The multidisciplinary team (involving also the clinician) decided after multiple consultations to start treatment directed at Sneathia sp as the condition of the patient was deteriorating, was not responding to current therapy and literature provided evidence for a potential role of Sneathia sp in the condition of the patient.

Lines 261-262: for S24, there was supporting evidence of *S. aureus* so the lab director could consider verbally informing the clinical team of the findings. However, for S45, samples from the surgical site should have been recollected and urogenital samples should have been tested for

N. gonorrhoeae to rule out disseminated gonococcal infection. Were there follow ups on these cases?

We agree with the reviewer that these results could be considered to communicate to the clinical team and added this to the revised manuscript (lines 286-287). Regarding S24, patient was treated for S. aureus endocarditis. Regarding S45, when RC-PCR was performed on the material the patient already recovered so no follow-up diagnostics were performed unfortunately targeted at N. gonorrhoeae.

Line 246: what exactly was used as a negative control?

These negative controls are samples that have been processed simultaneously with the clinical samples and processed using the full protocol from DNA isolation to RC-PCR but without adding material. This information has been added to the caption in Figure S1 (Supplementary information).

Line 266: the choice of reference databases is critical to approaches such as this one. Databases chosen should be well maintained and curated. Commercial products may be considered instead of NBCI which is quite disorganized and minimally curated. SILVA is good but has not been updated in a while. Once a database is updated, a mini verification of the new database should be performed to ensure that the new database does not affect the performance of the assay. Another thing to be mindful about is the constant changes in microbial nomenclature. The lab would need to be on top of these things and be aware of updates in the databases used for analysis and to make sure clinicians are aware of these changes.

The authors thank the reviewer for his thoughts on the reference database subject. Authors also agree on this topic and internally always verify and validate if a database is updated. This is standard procedure for our bioinformatics team and is always checked by our QC officer.

Lines 281: human DNA depletion followed by deep unbiased metagenomic sequencing may be a more appropriate method if the authors wanted to extend the scope of assay beyond organism identification.

Line 304-306: Added the line "Another appropriate but more expensive and less sensitive method would be to perform Human DNA depletion followed by metagenome sequencing referred to as clinical metagenomics (Chiu and Miller, Nature Reviews Genetics, 2019).

Line 335: The authors did not mention EasySeq, which is a commercial product, until the very end of the manuscript. Should the authors mention this sooner since an established commercial product was used throughout the entire project?

Authors of this paper co-developed, designed, and optimized the assay together with NimaGen B.V before release of the EasySeq product. Furthermore, authors designed the study and developed the bioinformatics independently from NimaGen B.V. Outcome of the assay was evaluated and written down without interference of NimaGen B.V. This has been added to the revised version, page 14.

We engaged this collaboration because we wanted to innovate our diagnostics and the RC technology.

Table 2 item S35: which serology are we talking about?

Anti-DNase B and ASO antibodies. We added this to Table 3.

Table 2 item S52: there really was no clinical info for this case? Dermacoccus could be a rare cause of catheter and shunt-associated infections. Did this patient have a VP shunt or any indwelling devices?

This case involves a patient from a different hospital from which we received material to perform 16S. For this reason, we unfortunately do not have any clinical information as the 16S Sanger result at that time remained negative so there was no consultation.

Supplemental info:

1. Did the authors see anything within reads that didn't assemble into 16S genes? This reviewer wondered how many human reads there were within the data. What measures did the authors have in place to scrub out human reads before the data are processed through the read classifier?

The bioinformatic analysis uses a read cleaning tool that removes low quality sequence reads. Almost no human reads have been sequenced and the method used for classifying the reads to the SILVA database selects the 16S targeting reads with certain quality so scrubbing out the human reads is redundant.

2. What were the average insert size for PCR products generated by 16S RC-PCR? The authors chose the use the 2x150 kit without tagmentation. Were the authors sure that all PCR products were sequenced in their entirety since the reads were so short? If the PCR product is larger than 300 bp the info in the middle may be missing. Why didn't the authors choose kits that provide longer reads and more data?

The reviewer is correct that we used 2x150. We used an Illumina MiniSeq. 1) for its faster turnaround time and 2) because this is also the workflow that we use for our SARS-CoV-2 mutation screening (Coolen et al., J Clin Virol, 2021). Using larger sequencing fragments would indeed cover the middle better and is therefore an approach that can be used but is not tested by us because of previous mentioned reasons.

May 9, 2023

Dr. Jordy P. M. Coolen
Radboudumc
Medical Microbiology
Geert Grooteplein Zuid 10
Nijmegen 6525 GA
Netherlands

Re: Spectrum04483-22R2 (Targeting 16S rRNA by reverse complement PCR-next-generation-sequencing: specific and sensitive detection and identification of microbes directly in clinical samples)

Dear Dr. Jordy P. M. Coolen:

Thank you for updating the manuscript per my previous review. Your manuscript has been accepted, and I am forwarding it to the ASM Journals Department for publication. You will be notified when your proofs are ready to be viewed.

Sincerely,

Paul Luethy
Editor, Microbiology Spectrum
